# SECOND-ORDER ALGORITHMS FOR FINDING LOCAL NASH EQUILIBRIA IN ZERO-SUM GAMES

## ABSTRACT

Zero-sum games arise in a wide variety of problems, including robust optimization and adversarial learning. However, algorithms deployed for finding a local Nash equilibrium in these games often converge to non-Nash stationary points. This highlights a key challenge: for any algorithm, the stability properties of its underlying dynamical system can cause non-Nash points to be potential attractors. To overcome this challenge, algorithms must account for subtleties involving the curvatures of players' costs. To this end, we leverage dynamical system theory and develop a second-order algorithm for finding a local Nash equilibrium in the smooth, possibly nonconvex-nonconcave, zero-sum game setting. First, we prove that this novel method guarantees convergence to only local Nash equilibria with a local *linear* convergence rate. We then interpret a version of this method as a modified Gauss-Newton algorithm with local *superlinear* convergence to the neighborhood of a point that satisfies first-order local Nash equilibrium conditions. In comparison, current related state-of-the-art methods do not offer convergence rate guarantees. Furthermore, we show that this approach naturally generalizes to settings with convex and potentially coupled constraints while retaining earlier guarantees of convergence to only local (generalized) Nash equilibria.

## 1 INTRODUCTION

We consider the setting of smooth, deterministic two-player zero-sum games of the form

$$\text{Player 1}: \min_{\mathbf{x}} f(\mathbf{x}, \mathbf{y}) \qquad \text{Player 2}: \max_{\mathbf{y}} f(\mathbf{x}, \mathbf{y}) \qquad (\mathbf{x}, \mathbf{y}) \in \mathcal{G}, \qquad \text{(Game 1)}$$

where $f$ can be nonconvex-nonconcave with respect to $\mathbf{x} \in \mathbb{R}^n$ and $\mathbf{y} \in \mathbb{R}^m$, respectively. In the unconstrained setting, i.e., when $\mathcal{G}$ is $(\mathbb{R}^n, \mathbb{R}^m)$, we seek to find a local Nash equilibrium. For the constrained setting, we will assume that $\mathcal{G}$ is convex and seek a local generalized Nash equilibrium.

Mathematical games are commonly studied in decision-making scenarios involving multiple agents in control theory (Isaacs, 1999), economics (Roth, 2002; Rubinstein, 1982), and computer science (Roughgarden, 2010). In particular, several problems of interest have a natural zero-sum game formulation, such as training generative adversarial networks (Goodfellow et al., 2014), pursuit-evasion scenarios (Isaacs, 1999), and robust optimization (Ben-Tal et al., 2009).

Several recent efforts (Fiez et al., 2020; Wang et al., 2020; Chinchilla et al., 2023; Daskalakis et al., 2023) consider a closely related minimax variant of Game 1, $\min_{\mathbf{x}} \max_{\mathbf{y}} f(\mathbf{x}, \mathbf{y})$; however, (local) *minimax* solutions can differ from (local) Nash equilibria in general nonconvex-nonconcave settings. This difference arises from the *order* of agent interactions. At a Nash solution of Game 1, players controlling $\mathbf{x}$ and $\mathbf{y}$ act *simultaneously*. In contrast, minimax points correspond to Stackelberg equilibria and assert a *sequential* order of play: $\mathbf{x}$ acts first, then $\mathbf{y}$ follows. We highlight this fact to point out that under the assumptions of Game 1, the set of all local Nash points is a subset of the set of all local Stackelberg points (Mazumdar et al., 2020; Ratliff et al., 2016). In particular, local Nash and Stackelberg points have the same first-order conditions but different second-order conditions.

The success of first-order gradient methods for single-agent learning problems made gradient descent ascent (GDA), the multi-agent analog of gradient descent, a natural starting point for solving Game 1. The GDA algorithm tries to find a critical point of $f$, i.e., where $\nabla f = \mathbf{0}$. GDA is known to get trapped in limit cycles even in the most straightforward convex-concave setting, and several works

have tried to modify the gradient dynamics by including second-order information to avoid this entrapment and direct the solution towards a stationary point of the dynamics (Benaım and Hirsch, 1999; Daskalakis et al., 2017; Hommes and Ochea, 2012; Mertikopoulos et al., 2018; Mescheder et al., 2017; Gidel et al., 2019). However, outside of the convex-concave setting, these methods can converge to critical points that are *not* Nash equilibria. This behavior is due to the particular structure of second-order derivatives of $f$ with respect to $\mathbf{x}$ and $\mathbf{y}$, and while they do not arise in the single-agent settings, they are widely documented in the multi-agent zero-sum game setting (Balduzzi et al., 2018; Mazumdar et al., 2020; Ratliff et al., 2016).

To guarantee that an algorithm only converges to local Nash equilibria, the algorithm's gradient dynamics must not have any non-Nash stable equilibrium points. To the best of our knowledge, only two previous works, local symplectic surgery (LSS) (Mazumdar et al., 2019) and curvature exploitation for the saddle point problem (CESP) (Adolphs et al., 2019), have such guarantees for the unconstrained nonconvex-nonconcave version of Game 1. However, neither of these methods provides any convergence rate analysis. Further, these works do not discuss the constrained setting of Game 1. A variety of Bregman proximal algorithms do find local min-max points in constrained, nonconvex-nonconcave settings with at best linear rates of convergence; however, they operate under the restrictive, blanket assumption that every critical point of $f$ is a local Nash equilibrium (Azizian et al., 2024), which is not generally true in nonconvex-nonconcave settings.

In this paper, we introduce second-order algorithms to solve Game 1. We highlight our specific contributions below:

1. We introduce **D**iscrete-time **N**ash **D**ynamics (`DND`), a discrete-time dynamical system that provably converges to only local Nash equilibria of the unconstrained version of Game 1 with a linear local convergence rate, while previous related work does not provide any convergence rates.

2. We modify this dynamical system and construct an algorithm, **Sec**ond **O**rder **N**ash **D**ynamics (`SecOND`), which can converge superlinearly to the neighborhood of a point that satisfies first-order local Nash conditions.

3. We discuss the constrained setting of Game 1, where $\mathcal{G}$ is a convex set. In this case, we use Euclidean projections to modify `DND` and develop an algorithm, **Se**cond-order **Co**nstrained **N**ash **D**ynamics (`SeCoND`), which finds a local generalized Nash Equilibrium point. In contrast, previous work either does not consider this constrained setting and/or is restricted to the convex-concave case.

## 2 PRELIMINARIES

Throughout this paper, $\mathbf{x} \in \mathbb{R}^n, \mathbf{y} \in \mathbb{R}^m$, and $\mathbf{z} = (\mathbf{x}^\top, \mathbf{y}^\top)^\top \in \mathbb{R}^{n+m}$.

### 2.1 GAME-THEORETIC CONCEPTS

**Definition 2.1. (Strict local Nash equilibrium)** A strategy $(\mathbf{x}^*, \mathbf{y}^*) \in \mathbb{R}^n \times \mathbb{R}^m$ is a strict local Nash equilibrium of Game 1, if

$$f(\mathbf{x}^*, \mathbf{y}) < f(\mathbf{x}^*, \mathbf{y}^*) < f(\mathbf{x}, \mathbf{y}^*), \tag{1}$$

for all $\mathbf{x}$ and $\mathbf{y}$ in feasible neighborhoods of $\mathbf{x}^*$ and $\mathbf{y}^*$ respectively.

Under the smoothness assumption of Game 1, defining first-order and second-order equilibrium conditions can help identify whether a point is a local Nash equilibrium (Ratliff et al., 2016). For the unconstrained setting, any point that satisfies the conditions below is said to be a differential Nash equilibrium and is guaranteed to be a strict local Nash equilibrium.

**Definition 2.2. (Sufficient conditions for strict local Nash equilibrium)** A strategy $(\mathbf{x}^*, \mathbf{y}^*) \in \mathbb{R}^n \times \mathbb{R}^m$ is a differential Nash equilibrium (and thus, a strict local Nash equilibrium) of Game 1 when $\mathcal{X}$ is $\mathbb{R}^n$ and $\mathcal{Y}$ is $\mathbb{R}^m$, if

$$\begin{aligned} \nabla_{\mathbf{x}} f(\mathbf{x}^*, \mathbf{y}^*) = 0, \quad & \nabla_{\mathbf{y}} f(\mathbf{x}^*, \mathbf{y}^*) = 0 \\ \nabla_{\mathbf{xx}}^2 f(\mathbf{x}^*, \mathbf{y}^*) \succ 0, \quad & \nabla_{\mathbf{yy}}^2 f(\mathbf{x}^*, \mathbf{y}^*) \prec 0. \end{aligned} \tag{2}$$

We now discuss the constrained version of Game 1. This paper allows the constrained setting to have coupled constraints. In the presence of coupling, the Nash equilibrium sought is a generalized Nash equilibrium.

**Definition 2.3. (Local generalized Nash equilibrium)** Assume the set $\mathcal{G}$ is convex. A strategy $(\mathbf{x}^*, \mathbf{y}^*) \in \mathcal{G}$ is a local generalized Nash equilibrium of Game 1 if

$$
\begin{aligned}
f(\mathbf{x}^*, \mathbf{y}^*) \leq f(\mathbf{x}, \mathbf{y}^*) \,\forall\, (\mathbf{x}, \mathbf{y}^*) \in \mathcal{G} \text{ in a neighborhood around } (\mathbf{x}^*, \mathbf{y}^*) \\
f(\mathbf{x}^*, \mathbf{y}^*) \geq f(\mathbf{x}^*, \mathbf{y}) \,\forall\, (\mathbf{x}^*, \mathbf{y}) \in \mathcal{G} \text{ in a neighborhood around } (\mathbf{x}^*, \mathbf{y}^*).
\end{aligned}
\tag{3}
$$

The optimality conditions of generalized Nash equilibria in the above-mentioned settings are well studied (Facchinei and Kanzow, 2010a;b). Though a standard treatment would involve defining the Karush-Kuhn-Tucker conditions for Game 1, for our purpose, the following conditions are sufficient for a point to be a local generalized Nash equilibrium.

**Definition 2.4. (Sufficient conditions for local generalized Nash equilibrium)** Assume the set $\mathcal{G}$ is convex. Let $\partial\mathcal{G}$ denote the set of boundary points of $\mathcal{G}$ and let $\mathcal{N}(\mathbf{x}, \mathbf{y})$ denote a neighbourhood around $(\mathbf{x}, \mathbf{y})$. Then:

- If for a strategy $(\mathbf{x}^*, \mathbf{y}^*) \in \mathcal{G}$,

$$
\nabla_{\mathbf{x}} f(\mathbf{x}^*, \mathbf{y}^*) = 0, \quad \nabla_{\mathbf{y}} f(\mathbf{x}^*, \mathbf{y}^*) = 0 \text{ and}
$$
$$
\nabla_{\mathbf{xx}}^2 f(\mathbf{x}^*, \mathbf{y}^*) \succ 0, \quad \nabla_{\mathbf{yy}}^2 f(\mathbf{x}^*, \mathbf{y}^*) \prec 0,
$$

  then $(\mathbf{x}^*, \mathbf{y}^*)$ is a *strict* local generalized Nash equilibrium of Game 1.

- If for a strategy $(\mathbf{x}^*, \mathbf{y}^*) \in \partial\mathcal{G}$

$$
\left( \begin{bmatrix} \mathbf{x} \\ \mathbf{y} \end{bmatrix} - \begin{bmatrix} \mathbf{x}^* \\ \mathbf{y}^* \end{bmatrix} \right)^{\top} \begin{bmatrix} \nabla_{\mathbf{x}} f(\mathbf{x}^*, \mathbf{y}^*) \\ -\nabla_{\mathbf{y}} f(\mathbf{x}^*, \mathbf{y}^*) \end{bmatrix} > 0 \,\forall\, (\mathbf{x}, \mathbf{y}) \in \mathcal{G}/(\mathbf{x}^*, \mathbf{y}^*) \cap \mathcal{N}(\mathbf{x}^*, \mathbf{y}^*)
$$

  then $(\mathbf{x}^*, \mathbf{y}^*)$ is a *strict* local generalized Nash equilibrium of Game 1. The strictness is lost if the inequality can hold with equality.

We now describe some concepts from dynamical system theory that determine whether an algorithm can converge to a local Nash equilibrium.

## 2.2 A Dynamical Systems Perspective

We illustrate how considerations of dynamical system theory are naturally motivated in our work through the example of GDA. We define:

$$
\omega(\mathbf{z}) := \begin{bmatrix} \nabla_{\mathbf{x}} f(\mathbf{x}, \mathbf{y}) \\ -\nabla_{\mathbf{y}} f(\mathbf{x}, \mathbf{y}) \end{bmatrix}, \quad J(\mathbf{z}) := \nabla_{\mathbf{z}} \omega(\mathbf{z}) = \begin{bmatrix} \nabla_{\mathbf{xx}}^2 f(\mathbf{x}, \mathbf{y}) & \nabla_{\mathbf{xy}}^2 f(\mathbf{x}, \mathbf{y}) \\ -\nabla_{\mathbf{yx}}^2 f(\mathbf{x}, \mathbf{y}) & -\nabla_{\mathbf{yy}}^2 f(\mathbf{x}, \mathbf{y}) \end{bmatrix}.
\tag{4}
$$

For some stepsize $\gamma$, the GDA update for Game 1 for any iteration $k$ can thus be written as

$$
\mathbf{z}_{k+1} = g_{\mathrm{GDA}}(\mathbf{z}_k) := \mathbf{z}_k - \gamma\omega(\mathbf{z}_k).
\tag{5}
$$

Equation (5) can be viewed as a discrete-time dynamical system. We may also consider the limiting ordinary differential equation of (5), obtained by taking infinitely small $\gamma$, which leads to a continuous-time dynamical system

$$
\dot{\mathbf{z}} = -\omega(\mathbf{z}).
\tag{6}
$$

Note that $-J(\mathbf{z})$ is the Jacobian of the continuous-time dynamical system in (6). We now introduce concepts we will build upon to comment on the behavior of any algorithm used to solve Game 1.

**Definition 2.5. (Critical point)** Given a continuous-time dynamical system $\dot{\mathbf{z}} = -h_c(\mathbf{z})$, $\mathbf{z} \in \mathbb{R}^{n+m}$ is a critical point of $h_c$ if $h_c(\mathbf{z}) = 0$. Further, if for a critical point $\mathbf{z}$, $\lambda \neq 0 \,\forall\, \lambda \in \mathrm{spec}(\nabla_{\mathbf{z}} h_c(\mathbf{z}))$, then $\mathbf{z}$ is called a hyperbolic critical point.

We can also define a similar concept for the discrete-time dynamical system counterpart.

**Definition 2.6. (Fixed point)** Given a discrete-time dynamical system $\mathbf{z}_{k+1} = h_d(\mathbf{z}_k), k \geq 0$, $\mathbf{z} \in \mathbb{R}^{n+m}$ is a fixed point of $h_d$ if $h_d(\mathbf{z}) = \mathbf{z}$.

Out of the various critical and fixed point types, we are interested in locally asymptotically stable equilibria (LASE) because they are the only locally exponentially attractive hyperbolic points under the dynamics flow. This means that any dynamical system that starts close enough to a LASE point will converge to that point.

**Definition 2.7. (Continuous-time LASE)** A critical point $\mathbf{z}^* \in \mathbb{R}^{n+m}$ of $h_c$ is a LASE of the continuous-time dynamics $\dot{\mathbf{z}} = -h_c(\mathbf{z})$ if $\mathrm{Re}(\lambda) > 0 \; \forall \; \lambda \in \mathrm{spec}(\nabla_{\mathbf{z}} h_c(\mathbf{z}^*))$.

**Definition 2.8. (Discrete-time LASE)** A fixed point $\mathbf{z}^* \in \mathbb{R}^{n+m}$ of $h_d$ is a LASE of the discrete-time dynamics $\mathbf{z}_{k+1} = h_d(\mathbf{z}_k), k \geq 0$ if $\rho(\nabla_{\mathbf{z}} h_d(\mathbf{z}^*)) < 1$, where $\rho(A)$ denotes the spectral radius of some matrix $A$.

### 2.3 MOTIVATION: LIMITING BEHAVIOR OF GDA

To motivate our work, we provide an overview of key results that analyze how GDA performs when applied to Game 1 (Balduzzi et al., 2018; Mazumdar et al., 2019; 2020). If GDA converges to a hyperbolic point $\mathbf{z}_{\mathrm{GDA}}$, GDA must have converged to a LASE. Thus, from definition 2.7,

$$\mathrm{Re}(\lambda) > 0 \; \forall \; \lambda \in \mathrm{spec}\left( \underbrace{\begin{bmatrix} \nabla_{\mathbf{xx}}^2 f(\mathbf{x}_{\mathrm{GDA}}, \mathbf{y}_{\mathrm{GDA}}) & \nabla_{\mathbf{xy}}^2 f(\mathbf{x}_{\mathrm{GDA}}, \mathbf{y}_{\mathrm{GDA}}) \\ -\nabla_{\mathbf{yx}}^2 f(\mathbf{x}_{\mathrm{GDA}}, \mathbf{y}_{\mathrm{GDA}}) & -\nabla_{\mathbf{yy}}^2 f(\mathbf{x}_{\mathrm{GDA}}, \mathbf{y}_{\mathrm{GDA}}) \end{bmatrix}}_{J(\mathbf{z}_{\mathrm{GDA}})} \right). \tag{7}$$

Clearly, if $\mathbf{z}_{\mathrm{GDA}}$ happens to be a strict local Nash equilibrium, from (4), we know that $\nabla_{\mathbf{xx}}^2 f(\mathbf{x}_{\mathrm{GDA}}, \mathbf{y}_{\mathrm{GDA}}) \succ 0$ and $\nabla_{\mathbf{yy}}^2 f(\mathbf{x}_{\mathrm{GDA}}, \mathbf{y}_{\mathrm{GDA}}) \prec 0$. Hence, from definition 2.7, it is clear that *all* strict local Nash equilibria of Game 1 are LASE of the GDA dynamics. However, the converse cannot be guaranteed, and thus, a LASE point to which GDA converges may *not* be a local Nash equilibrium.

Let us further examine the structure of $J$:

$$J(\mathbf{z}) = \begin{bmatrix} A & B \\ -B^\top & D \end{bmatrix}, \forall \; \mathbf{z} \in \mathbb{R}^{n+m}. \tag{8}$$

Only two previous works, LSS (Mazumdar et al., 2019) and CESP (Adolphs et al., 2019), leverage this structure and propose dynamical systems that have *only* local Nash equilibria as their LASE. However, the convergence rates of these methods have not been analyzed. Further, neither of these methods discusses the constrained case, which arises in many practical situations.

This motivates us to develop a novel second-order method with a dynamical system that guarantees that only local Nash equilibria constitute its LASE points, generalizes to the constrained settings, and has an established convergence rate.

## 3 OUR METHOD AND MAIN RESULTS

We are now ready to show our main results. We begin with the unconstrained setting and then move to the constrained setting. All proofs are given in Appendix A.

### 3.1 UNCONSTRAINED SETTING

We list the common assumptions we make for the entire unconstrained case below, and we discuss their validity in Appendix B.

**Assumption 1.** The objective function $f \in \mathcal{C}^3$.

**Assumption 2.** $J(\mathbf{z}), \nabla_{\mathbf{xx}}^2 f(\mathbf{x}, \mathbf{y}), \nabla_{\mathbf{yy}}^2 f(\mathbf{x}, \mathbf{y})$ are invertible at all $\mathbf{z}$ where $\omega(\mathbf{z}) = 0$.

**Assumption 3.** $\omega(\mathbf{z})$ does not belong to the null space of $J(\mathbf{z})^\top$, for all $\mathbf{z} \in \mathbb{R}^{n+m}$.

**Assumption 4.** $\omega$ is $L_\omega$-Lipschitz, and $J$ is $L_J$-Lipschitz.

**Motivation.** We first introduce a continuous-time dynamical system that employs second-order derivative information, for which we can establish desirable properties and which motivates our main method. Consider the system:

$$\dot{\mathbf{z}} = -g_c(\mathbf{z}) = -\left[ J(\mathbf{z})^\top J(\mathbf{z}) \left( J(\mathbf{z}) + J(\mathbf{z})^\top \right) + E_c(\mathbf{z}) \right]^{-1} J(\mathbf{z})^\top \omega(\mathbf{z}), \tag{9}$$

where $E_c(\mathbf{z})$ is a regularization matrix chosen such that $J(\mathbf{z})^\top J(\mathbf{z}) \left( J(\mathbf{z}) + J(\mathbf{z})^\top \right) + E_c(\mathbf{z})$ is invertible, and $\omega(\mathbf{z}) = 0 \implies E_c(\mathbf{z}) = 0$. Under assumptions 1, 2, 3, and 4 all solutions of (9) converge to a strict local Nash equilibrium in the unconstrained setting of Game 1. This is because strict local Nash equilibria of Game 1 are the *only* LASE points of (9). To prove this, we first show that critical points of $g_c$ and $\omega(\mathbf{z})$ are the same.

**Lemma 1.** *Under Assumptions 1, 2, 3, and 4, the critical points of $g_c$ are exactly the critical points of the GDA dynamics $\dot{\mathbf{z}} = -\omega(\mathbf{z})$.*

Lemma 1 establishes that at every LASE $\mathbf{z}$ of (9), $\omega(\mathbf{z}) = 0$. This helps us to prove that (9) converges to only a strict local Nash equilibrium.

**Theorem 1.** *Under Assumptions 1, 2, 3, and 4, $\mathbf{z}$ is a LASE point of $\dot{\mathbf{z}} = -g_c(\mathbf{z})$ if and only if $\mathbf{z}$ is a strict local Nash equilibrium of Game 1.*

**Remark 1. (Avoiding rotational instability)** *It is well documented that oscillations around equilibria are caused if the Jacobian of the gradient dynamics has eigenvalues with dominant imaginary parts near equilibria (Mescheder et al., 2017; Balduzzi et al., 2018; Gidel et al., 2019; Mazumdar et al., 2019; Wang et al., 2020). Corollary 1 establishes that this cannot happen for the dynamics (9).*

**Corollary 1.** *Under Assumptions 1, 2, 3, and 4, if $\mathbf{z}$ is a strict local Nash equilbrium of $g_c$, then the Jacobian $\nabla g_c$ has only real eigenvalues at $\mathbf{z}$.*

**Practical Considerations.** Although the continuous-time dynamical system we introduce in (9) has desirable theoretical properties, it is not yet a practical algorithm that can solve Game 1. To solve Game 1, we require a discrete-time dynamical system. Inspired from (9), we propose **D**iscrete-time **N**ash **D**ynamics (DND):

$$
\begin{aligned}
\mathbf{z}_{k+1} &= g_d(\mathbf{z}_k) \\
&= \mathbf{z}_k - \alpha_k \left( \left[ J(\mathbf{z}_k)^\top J(\mathbf{z}_k) \left( J(\mathbf{z}_k) + J(\mathbf{z}_k)^\top + \beta(\mathbf{z}_k) \right) + E(\mathbf{z}_k) \right]^{-1} \right) J(\mathbf{z}_k)^\top \omega(\mathbf{z}_k).
\end{aligned} \tag{10}
$$

Regularization $E(\mathbf{z}_k)$ is chosen to maintain invertibility in (10) and adheres to the condition that $\omega(\mathbf{z}_k) = 0 \implies E(\mathbf{z}_k) = 0$. In contrast to the continuous-time system $g_c$ in (9), DND in (10) contains an extra regularization term $\beta(\mathbf{z}_k)$. Adding $\beta(\mathbf{z}_k)$ guarantees the stability of (10) in accordance with definition 2.8, and is given by

$$
\beta(\mathbf{z}) = \begin{bmatrix} \mathbb{1}_{\{\lambda_{\mathbf{x}} > 0\}}(b_{\mathbf{x}})I & 0 \\ 0 & \mathbb{1}_{\{\lambda_{\mathbf{y}} < 0\}}(b_{\mathbf{y}})I \end{bmatrix}, \tag{11}
$$

where $\lambda_{\mathbf{x}}$ and $\lambda_{\mathbf{y}}$ denote the minimum and maximum eigenvalues of $\nabla^2_{\mathbf{xx}} f(\mathbf{x}, \mathbf{y})$ and $\nabla^2_{\mathbf{yy}} f(\mathbf{x}, \mathbf{y})$ respectively. These eigenvalues can be found through computations involving Hessian-vector products, which can be made as efficient as gradient evaluations (Pearlmutter, 1994; Lanczos, 1950). The terms $b_{\mathbf{x}}$ and $b_{\mathbf{y}}$ can be taken to be any constants as long as $b_{\mathbf{x}} > 1/2$ and $b_{\mathbf{y}} < -1/2$.

$\beta(\mathbf{z})$ is a non-smooth regularization term, but it is differentiable around any fixed point of $\omega$. The following theorem shows that DND inherits all the desirable properties that we established for the continuous-time system $g_c$.

**Theorem 2.** *Under Assumptions 1, 2, 3, and 4, for any $\alpha_k \in (0, 1]$, DND, with $\beta(\mathbf{z})$ chosen as in (11) satisfies the following:*

1. *The fixed points of DND are exactly the fixed points of the discrete-time GDA dynamics in (5).*

2. *$\mathbf{z}$ is a LASE of DND $\iff$ $\mathbf{z}$ is a strict local Nash equilibrium of unconstrained Game 1.*

3. *If $\mathbf{z}$ is a fixed point of DND, then the Jacobian $\nabla g_d$ has only real eigenvalues at $\mathbf{z}$.*

We find that DND has a *linear* local convergence rate.

**Theorem 3.** *Assume that a strict local Nash equilibrium of Game 1 exists. Under Assumptions 1, 2, 3, and 4, if DND converges, it converges to a strict local Nash equilibrium of Game 1. Further, if the step size is chosen as $\alpha_k \leq \max\{2|\lambda_{\mathbf{x}}|, 2|\lambda_{\mathbf{y}}|\}$ then DND has a linear local convergence rate of*

$$
\lim_{k \to \infty} \frac{\|\mathbf{z}_{k+1} - \mathbf{z}^*\|}{\|\mathbf{z}_k - \mathbf{z}^*\|} \leq \max \left\{ \left( 1 - \frac{\alpha}{2\tilde{\lambda}_{\mathbf{x}}} \right), \left( 1 + \frac{\alpha}{2\tilde{\lambda}_{\mathbf{y}}} \right) \right\}.
$$

*Here, $\alpha$ is the step size at the sequence limit in (10), and $\lambda_{\mathbf{x}}, \lambda_{\mathbf{y}}$ refer to the quantities in (11), and $\tilde{\lambda}_{\mathbf{x}} > 0, \tilde{\lambda}_{\mathbf{x}} < 0$ denote $\lambda_{\mathbf{x}}, \lambda_{\mathbf{y}}$ evaluated at the sequence limit.*

**Can we speed up `DND`?** We motivate a modification to (10), which allows for superlinear convergence to a ball-shaped region around a fixed point. If this fixed point is a LASE (and therefore also a local Nash equilibrium), the modification can achieve rapid convergence to a small neighborhood of this local Nash point. The modification retains desirable stability guarantees and escapes the ball if the fixed point is not a LASE. The radius of the ball can be treated as a hyperparameter and tuned for good performance.

**Modified discrete-time system.** We call the modified method **Sec**ond **O**rder **N**ash **D**ynamics (`SecOND`), which is given by

$$\mathbf{z}_{k+1} = \begin{cases} z_k - \alpha_k \left(S(\mathbf{z}_k)\right)^{-1} J(\mathbf{z}_k)^\top \omega(\mathbf{z}_k), & \|\mathbf{z}_k - \mathbf{z}_{k-1}\| > \epsilon \\ g_d(\mathbf{z}_k), & \text{else.} \end{cases} \quad (12)$$

where $\epsilon > 0$ is a user-specified constant, the matrix $S(\mathbf{z}_k) \succ 0$ and can be derived from modifying the off-diagonal terms of $J(\mathbf{z}_k)^\top J(\mathbf{z}_k) \left(J(\mathbf{z}_k) + J(\mathbf{z}_k)^\top + \beta(\mathbf{z})\right)$ with an appropriate $E(\mathbf{z}_k)$ in (10). We define such a choice in Appendix D.

**Reinterpretation as a Gauss-Newton method far from fixed points.** Consider the problem

$$\min_{\mathbf{z}\in\mathbb{R}^{n+m}} \underbrace{\frac{1}{2}\|\omega(\mathbf{z})\|_2^2}_{l(\mathbf{z})}. \quad (13)$$

We observe that $\nabla_{\mathbf{z}} l(\mathbf{z}) = J(\mathbf{z})^\top \omega(\mathbf{z})$. For $\|\mathbf{z}_k - \mathbf{z}_{k-1}\| > \epsilon$, we have the system $\mathbf{z}_{k+1} = \mathbf{z}_k - (S(\mathbf{z}_k))^{-1} \nabla_{\mathbf{z}} l(\mathbf{z})$, with $S(\mathbf{z}_k) \succ 0$, which is a *modified Gauss-Newton* algorithm for solving (13). By choosing $S(\mathbf{z}_k) \approx J(\mathbf{z}_k)^\top J(\mathbf{z}_k)$ (see Appendix D), if the Gauss-Newton system converges to a fixed point $\mathbf{z}_c$, we can be guaranteed a *superlinear rate of convergence to that point*. Moreover, whenever $\|\mathbf{z}_k - \mathbf{z}_{k-1}\| > \epsilon$, we may choose step size $\alpha_k$ according to any standard line search rule from nonlinear programming (Nocedal and Wright, 1999; Bertsekas, 1997). For example, in our implementation, we choose a backtracking line search with the Armijo condition (Armijo, 1966) and choose an $\alpha_k$ for some $c \in (0, 1)$ such that

$$l(\mathbf{z}_k) - l(\mathbf{z}_{k+1}) \geq c\alpha_k \omega(\mathbf{z}_k)^\top J(\mathbf{z}_k)(S(\mathbf{z}_k))^{-1} J(\mathbf{z}_k)^\top \omega(\mathbf{z}_k). \quad (14)$$

Based on `SecOND`, we construct Algorithm 1, which converges superlinearly toward the first critical point it encounters, and switches to `DND` when it is close enough to that point. If the critical point satisfies the strict local Nash equilibrium sufficiency conditions given in Definition (2.2), `SecOND` will have reached the point faster than `DND` would have from the same initialization. If the fixed point does not satisfy strict local Nash conditions, the switch to `DND` dynamics ensures that iterates escape, and avoids convergence to the spurious fixed point. More sophisticated variants which allow for switching back and forth multiple times can also be considered.

**Theorem 4.** *Under Assumptions 1, 2, 3, and 4, $\mathbf{z}$ is a LASE of `SecOND` (Algorithm 1) if and only if $\mathbf{z}$ is a strict local Nash equilibrium of Game 1. Further, assume that a strict local Nash equilibrium of Game 1 exists and let $\mathbf{z}_c$ be the first critical point to which Algorithm 1 comes near to. If $S(\mathbf{z}_k) \approx J(\mathbf{z}_k)^\top J(\mathbf{z}_k)$ (see Appendix D), then Algorithm 1 approaches $\mathbf{z}_c$ superlinearly with a rate*

$$\|\mathbf{z}_{k+1} - \mathbf{z}_c\| \leq L_\omega L_J M\|\mathbf{z}_k - \mathbf{z}_c\|^2, \forall\, k = 0, 1, \dots.$$

*Here $M = \sup_{z\in\tilde{\mathcal{B}}}\|S(\mathbf{z})^{-1}\|$, where $\tilde{\mathcal{B}}$ is the smallest ball centered at $\mathbf{z}_c$ which contains $\mathbf{z}_0$.*

*Furthermore, if Algorithm 1 converges, it converges to a strict local Nash equilibrium of Game 1.*

Theorem 4 establishes that `SecOND` inherits the desirable stability properties of `DND`, while being faster than `DND` in approaching a critical point.

---

**Algorithm 1 Sec**ond **O**rder **N**ash **D**ynamics (SecOND)

---

**Input:** Functions $\omega(\mathbf{z}), J(\mathbf{z}), S(\mathbf{z})$; initial point $\hat{\mathbf{z}}$; constants $\epsilon > 0, 0 < \alpha_0 \leq 1$
**Initialize:** $\mathbf{z}_0 \leftarrow \hat{\mathbf{z}}, \mathbf{z}_1 \leftarrow \mathbf{z}_0 - \alpha_0 (S(\mathbf{z}_0))^{-1} J(\mathbf{z}_0)^\top \omega(\mathbf{z}_0), k = 1$
    **while** not converged **do**
        **if** $\|\mathbf{z}_k - \mathbf{z}_{k-1}\| > \epsilon$ **then**
            Choose $\alpha_k$ with appropriate line search          $\triangleright$ for example, from (14)
            $\mathbf{z}_{k+1} \leftarrow z_k - \alpha_k (S(\mathbf{z}_k))^{-1} J(\mathbf{z}_k)^\top \omega(\mathbf{z}_k)$          $\triangleright$ from (12)
        **else if** $\mathbf{z}_k$ does not satisfy strict LNE sufficiency conditions **then**      $\triangleright$ from Definition (2.2)
            $\mathbf{z}_{k+1} \leftarrow g_d(\mathbf{z}_k)$          $\triangleright$ from (10)
        **else**
            **break**
        **end if**
        $k \leftarrow k + 1$
    **end while**
    **return** $\mathbf{z}_k$

---

## 3.2 CONSTRAINED SETTING

**Notation.** $\Pi_{\mathcal{Q}}[\mathbf{p}]$ denotes the Euclidean projection of some vector $\mathbf{p}$ onto some set $\mathcal{Q}$. $\mathrm{proj}_{\mathbf{a}}(\mathbf{b})$ denotes the Euclidean projection of a vector $\mathbf{b}$ onto another vector $\mathbf{a}$. $\mathrm{int}\,\mathcal{G}$ and $\partial\mathcal{G}$ denote the interior and boundary of $\mathcal{G}$ respectively.

Intuitively, any local generalized Nash equilibrium in $\mathrm{int}\,\mathcal{G}$ is actually also a strict local Nash equilibrium of the unconstrained game. Therefore, if the Euclidean projections of the DND iterates converge to a point in $\mathrm{int}\,\mathcal{G}$, this point must be a local generalized Nash equilibrium. Further, if a step taken by DND at a point $\mathbf{z}$ in $\partial\mathcal{G}$ is parallel to $-\omega(\mathbf{z})$, then, from definition 2.4, $\mathbf{z}$ is a local Generalized Nash equilibrium as well.

**Algorithm for Constrained Setting.** Based on the above discussion, we construct **Se**cond-order **Co**nstrained **N**ash **D**ynamics (SeCoND), given in Algorithm 2, for solving a constrained Game 1. SeCoND has the property that if it converges, it converges to a local Generalized Nash equilibrium that follows definition 2.4. If desired, Algorithm 2 convergence can be accelerated via a Gauss-Newton approach analogous to (12).

**Assumption 5.** The set $\mathcal{G}$ is convex.

**Theorem 5.** *Let Assumptions 1, 2, 3, 4, and 5 hold, and let $\omega(\mathbf{z}) \neq 0 \,\forall\, \mathbf{z} \in \partial G$. Then, if SeCoND (Algorithm 2) converges to a point $\mathbf{z}$:*

    *1. If $\mathbf{z} \in \mathrm{int}G$, then $\mathbf{z}$ is a strict local generalized Nash equilibrium.*

    *2. If $\mathbf{z} \in \partial G$, then $\mathbf{z}$ is a local generalized Nash equilibrium (not necessarily strict).*

---

**Algorithm 2 Se**cond-order **C**onstrained **N**ash **D**ynamics (SeCoND)

---

**Input:** Functions $\omega(\mathbf{z}), J(\mathbf{z})$; set $\mathcal{G}$; initial point $\hat{\mathbf{z}}$; constant $\alpha$
**Initialize:** $\mathbf{z}_0 \leftarrow \hat{\mathbf{z}}, k = 0$
    **while** not converged **do**
        **if** $\mathbf{z}_k \in \mathrm{Int}\,\mathcal{G}$ **then**
            $\mathbf{z}_{k+1} \leftarrow \Pi_{\mathcal{G}}\left[g_d(\mathbf{z}_k)\right]$          $\triangleright$ from (10)
        **else if** $\mathbf{z}_k \in \partial\mathcal{G}$ **then**          $\triangleright$ $E$ from (10), $\beta$ from (11)
            $\mathbf{m} \leftarrow \mathrm{proj}_{\omega(\mathbf{z}_k)}\left( \left[ J(\mathbf{z}_k)^\top J(\mathbf{z}_k) \left(J(\mathbf{z}_k) + J(\mathbf{z}_k)^\top + \beta(\mathbf{z}_k)\right) + E(\mathbf{z}_k)\right]^{-1} J(\mathbf{z}_k)^\top \omega(\mathbf{z}_k) \right)$
            $\mathbf{z}_{k+1} \leftarrow \Pi_{\mathcal{G}}\left[\mathbf{z}_k - \alpha\mathbf{m}\right]$
        **end if**
        $k \leftarrow k + 1$
    **end while**
    **return** $\mathbf{z}_k$

---

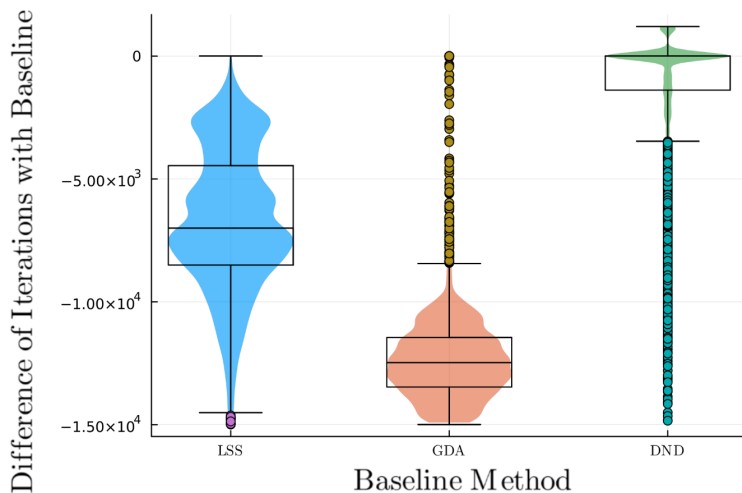

(a) Violin plot of the difference in iterations taken between `SecOND` and each baseline method (lower is better). `SecOND` converges faster than baselines for the unconstrained Game 1. Dots represent outliers, (see Appendix E.1.2).

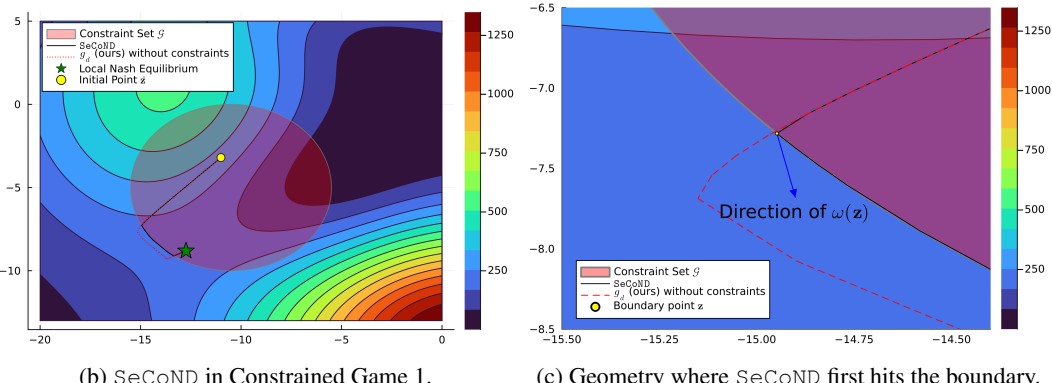

(b) `SeCoND` in Constrained Game 1.  (c) Geometry where `SeCoND` first hits the boundary.

Figure 1: Numerical results for a two-dimensional toy example.

## 4 EXPERIMENTS

We now investigate how well the theoretical properties of our algorithms transfer to practical problems. Our main aims are: (i) to compare the performance of `SecOND` with previous related work in unconstrained, nonconvex-nonconcave settings, (ii) to determine if modifications made to `DND` in `SecOND` are beneficial, (iii) to test whether `SeCoND` converges to a local generalized Nash equilibrium in the constrained setting. All details of the experimental setup are included in Appendix E.

### 4.1 TWO-DIMENSIONAL TOY EXAMPLE

We consider the function

$$f(x, y) = e^{-0.01(x^2+y^2)}((0.3x^2 + y)^2 + (0.5y^2 + x)^2), \; x, y \in \mathbb{R}.$$

This function is nonconvex-nonconcave, and the unconstrained version of Game 1 has three local Nash equilibria, while the GDA dynamical system (6) has 4 LASE points for this function.

**Baselines.** In this experiment, we tested the performance of `SecOND` (Algorithm 1) against three baselines: `DND`, gradient descent-ascent (GDA), and local symplectic surgery (LSS) (Mazumdar et al., 2019), on 10000 random initializations.

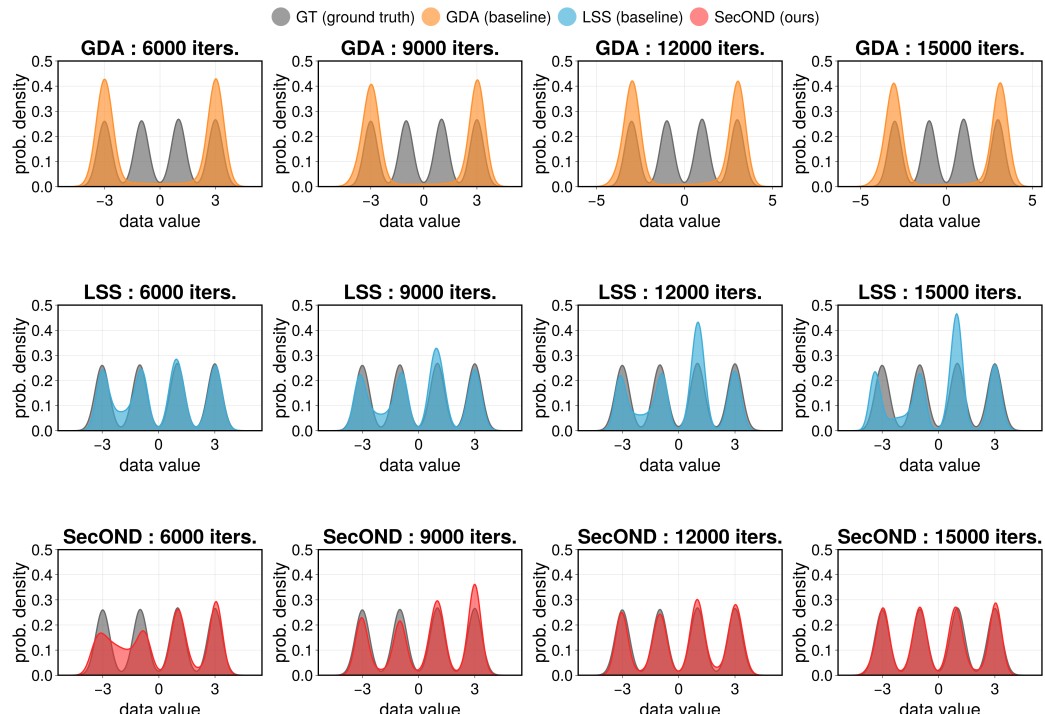

Figure 2: `SecOND` converges rapidly and to a more accurate solution for a GAN training problem.

**Does `SecOND` provide faster convergence than baselines?** Figure 1(a) shows the difference in the number of iterations taken to converge within a fixed tolerance by `SecOND` and each respective baseline. `SecOND` consistently converged more rapidly than LSS, achieving a still greater performance improvement than GDA. Finally, we note that we could not compare to the CESP method (Adolphs et al., 2019), because it could not reliably converge in our experiments (see Appendix E.1.1). An additional experiment investigating convergence of all algorithms to a local Nash equilibrium is in Appendix E.1.3.

**Does `SecOND` perform better than `DND`?** From Figure 1(a), we observe that `SecOND` performed similarly to `DND` in this numerical example. `DND` outperformed `SecOND` in some instances, which occurred when `SecOND` initially went to the neighborhood of an undesirable critical point, at which the quantity $\|\omega(\mathbf{z})\|_2^2 \approx 0$. In such cases, `SecOND` had to correct its course to go to the desirable fixed points. This made it converge slower than `DND`, which went to the desirable fixed points in the first place. In the cases when `SecOND` rapidly approaches a desirable critical point, `SecOND` converged much faster than `DND`. This shows that the modification made to `DND` in `SecOND` can indeed be advantageous.

**Does `SeCoND` converge to a local generalized Nash equilibrium?** We tested `SeCoND` (Algorithm 2) in this toy setting by including a constraint of the form $(x + 10.5)^2 + (y + 5)^2 \le 25$, and found that `SeCoND` successfully converges to a local generalized Nash equilibrium. As seen in Figure 1(b), `SeCoND` initially follows `DND` while iterates remain in the interior of the feasible set. However, after hitting the boundary, `SeCoND` remains on the boundary before returning to the interior and converging to the same local (generalized) Nash equilibrium as `DND`. Figure 1(c) is representative of the geometry across the portion where `SeCoND` remains on the boundary. Because $-\omega(\mathbf{z})$ is not parallel to the constraint gradient here, `SeCoND` eventually returns to the interior.

### 4.2 GENERATIVE ADVERSARIAL NETWORK (GAN)

Next, we consider a larger-scale test problem in which $\omega(\mathbf{z})$ is computed stochastically (i.e., via sampling minibatches of data). To this end, we evaluated GDA, LSS, and `SecOND` on a GAN training

problem where the generator must fit a 1D mixture of Gaussians with 4 mixture components. The distribution that each algorithm learned at different training iterations is plotted in Figure 2. GDA suffered mode collapse early on and only fit two out of the four modes. Both LSS and `SecOND` successfully found all four modes of the problem. While LSS initially seems to converge rapidly, continued training degrades performance. Over time, `SecOND` outperformed LSS and fit the ground truth distribution more closely by 12000 iterations.

## 5 CONCLUSION, LIMITATIONS, AND FUTURE WORK

We have provided algorithms that provably converge to only local Nash equilibria in smooth, possibly nonconvex-nonconcave, two-player zero-sum games in the unconstrained (`DND`, `SecOND`) and convex-constrained (`SeCoND`) settings. We have shown that `DND` has a *linear* local convergence rate and that `SecOND` approaches a neighborhood around a fixed point superlinearly. In contrast, the most closely related existing approaches for this setting have no established convergence rates. Empirical results demonstrate that `DND` and `SecOND` outperformed previous related works in two test problems.

**Limitations and Future Work.**   There are three key limitations of the proposed method. (i) All approaches in this paper require second-order information, which can be prohibitively expensive to obtain or compute in high-dimensional scenarios. Unfortunately, the fundamental links this problem shares with dynamical system theory necessitate second-order information to provide convergence guarantees. (ii) Like other approaches (Mazumdar et al., 2019), we require Assumption 3 in order to ensure that the critical points of the dynamics we introduce coincide with first-order local Nash points. Finally, (iii) as in other work on zero-sum Nash games (Adolphs et al., 2019; Mazumdar et al., 2019; 2020), we can only provide local convergence analysis, and cannot ensure that the dynamics globally converge (even if local Nash points do exist). Addressing this limitation is a key direction of future work. Future work should also aim to relax Assumption 3, and use algorithms introduced in this paper as building blocks for solving *dynamic* zero-sum games with simultaneous decision-making occurring across multiple time stages.

### REPRODUCIBILITY STATEMENT

For all theoretical analyses in this work, all assumptions can be found in Section 3, and each theorem/lemma/corollary states the particular assumptions involved. For all experimental details and parameter values, please refer to Appendix D and Appendix E. The code to reproduce the experiments, along with instructions to run them, is included in a supplementary zip submission.

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

# Appendix

## Table of Contents

## A  PROOFS

**Lemma 1.** *Under Assumptions 1, 2, 3, and 4, the critical points of $g_c$ are exactly the critical points of the GDA dynamics $\dot{\mathbf{z}} = -\omega(\mathbf{z})$.*

*Proof.* ( $\implies$ ) Clearly,
$$\omega(\mathbf{z}) = 0 \implies g_c(\mathbf{z}) = 0.$$
( $\impliedby$ ) Now assume that $\mathbf{z}$ is a critical point of $g_c$ such that $\omega(\mathbf{z}) \neq 0$. In this case, due to the choice of our regularization $E_c(\mathbf{z})$, $g_c(\mathbf{z})$ can be thought of as $g_c(\mathbf{z}) = M(\mathbf{z})\omega(\mathbf{z})$, where $M(\mathbf{z})$ is full rank. Thus,
$$g_c(\mathbf{z}) = 0 \implies M(z)\omega(\mathbf{z}) = 0 \implies \omega(\mathbf{z}) = 0,$$
which is a contradiction. Hence, $g_c(\mathbf{z}) = 0 \iff \omega(\mathbf{z}) = 0$. $\qquad\square$

**Theorem 1.** *Under Assumptions 1, 2, 3, and 4, $\mathbf{z}$ is a LASE point of $\dot{\mathbf{z}} = -g_c(\mathbf{z})$ if and only if $\mathbf{z}$ is a strict local Nash equilibrium of Game 1.*

*Proof.* ( $\implies$ ) As all LASE points of continuous-time dynamics are also critical points, for any LASE point $\mathbf{z} = (\mathbf{x}^\top, \mathbf{y}^\top)^\top$, $\omega(\mathbf{z}) = 0$. Thus the Jacobian of $g_c$ at $\mathbf{z}$ becomes

$$\nabla g_c(\mathbf{z}) = \left[ J(\mathbf{z})^\top J(\mathbf{z})(J(\mathbf{z}) + J(\mathbf{z})^\top) \right]^{-1} J(\mathbf{z})^\top J(\mathbf{z}) = (J(\mathbf{z}) + J(\mathbf{z})^\top)^{-1}$$
$$= H(\mathbf{z}) := \begin{bmatrix} \frac{1}{2} \left( \nabla^2_{\mathbf{xx}} f(\mathbf{x}, \mathbf{y}) \right)^{-1} & 0 \\ 0 & -\frac{1}{2} \left( \nabla^2_{\mathbf{yy}} f(\mathbf{x}, \mathbf{y}) \right)^{-1} \end{bmatrix}. \tag{15}$$

From definition 2.7,
$$\nabla g_c(\mathbf{z}) = H(\mathbf{z}) \succ 0 \implies \nabla^2_{\mathbf{xx}} f(\mathbf{x}, \mathbf{y}) \succ 0 \text{ and } \nabla^2_{\mathbf{yy}} f(\mathbf{x}, \mathbf{y}) \prec 0,$$

which implies that $(\mathbf{x}, \mathbf{y})$ is a strict local Nash equilibrium of Game 1 (from definition 2.2). Thus, every LASE of $\dot{\mathbf{z}} = -g_c(\mathbf{z})$ is a strict local Nash equilibrium of (Game 1).
( $\impliedby$ ) Consider a strict local Nash equilibrium $(\mathbf{x}^*, \mathbf{y}^*)$ of Game 1. From definition 2.2, $\nabla^2_{\mathbf{xx}} f(\mathbf{x}^*, \mathbf{y}^*) \succ 0, \nabla^2_{\mathbf{yy}} f(\mathbf{x}^*, \mathbf{y}^*) \prec 0$, and $\omega(\mathbf{z}^*) = 0$ where $\mathbf{z}^* = (\mathbf{x}^{*\top}, \mathbf{y}^{*\top})^\top$. Clearly, $H(\mathbf{z}^*) \succ 0$ and thus $\mathbf{z}^*$ is a LASE of (9). $\qquad\square$

**Corollary 1** *Under Assumptions 1, 2, 3, and 4, if $\mathbf{z}$ is a strict local Nash equilbrium of $g_c$, then the Jacobian $\nabla g_c$ has only real eigenvalues at $\mathbf{z}$.*

*Proof.* From theorem 1, $\mathbf{z}$ must also be a LASE, and by extension, a critical point of $g_c$. From lemma 1, $\omega(\mathbf{z}) = 0$. Consider (15). As the inverse Hessians $\left(\nabla^2_{\mathbf{xx}}f(\mathbf{x}, \mathbf{y})\right)^{-1}$ and $\left(\nabla^2_{\mathbf{yy}}f(\mathbf{x}, \mathbf{y})\right)^{-1}$ are symmetric, $H(\mathbf{z})$ is symmetric. Because $\omega(\mathbf{z}) = 0$, the Jacobian $\nabla g_c(\mathbf{z}) = H(\mathbf{z})$, and $H(\mathbf{z})$ only has real eigenvalues due to symmetry. $\qquad\square$

**Theorem 2.** *Under Assumptions 1, 2, 3, and 4, for any $\alpha_k \in (0, 1]$, DND, with $\beta(\mathbf{z})$ chosen as in (11) satisfies:*

1. *The fixed points of DND are exactly the fixed points of the discrete-time GDA dynamics in (5).*

2. *$\mathbf{z}$ is a LASE point of DND $\iff$ $\mathbf{z}$ is a strict local Nash equilibrium of unconstrained Game 1.*

3. *If $\mathbf{z}$ is a fixed point of DND, then the Jacobian $\nabla g_d$ has only real eigenvalues at $\mathbf{z}$.*

*Proof.* $(1. \implies)$ The fixed points of the discrete GDA dynamics in (5) are critical points of $\omega$, i.e, where $\omega(\mathbf{z}) = 0$. Clearly,

$$\omega(\mathbf{z}) = 0 \implies g_d(\mathbf{z}) = \mathbf{z}.$$

$(1. \impliedby)$ Now assume that $\mathbf{z}$ is a fixed point of $g_d$ such that $\omega(\mathbf{z}) \neq 0$. In this case, due to the choice of our regularization $E(\mathbf{z})$, $g_d(\mathbf{z})$ can be thought of as $g_d(\mathbf{z}) = \mathbf{z} - \alpha M(\mathbf{z})J(\mathbf{z})^\top \omega(\mathbf{z})$, where $M(\mathbf{z})$ is full rank and $\alpha$ is the step size. Thus,

$$g_d(\mathbf{z}) = \mathbf{z} \implies M(z)J(\mathbf{z})^\top \omega(\mathbf{z}) = 0 \implies \omega(\mathbf{z}) = 0,$$

which is a contradiction. Hence, $g_d(\mathbf{z}) = \mathbf{z} \iff \omega(\mathbf{z}) = 0$.

$(2. \implies)$ As all LASE points of discrete-time dynamics are also fixed points, for any LASE point $\mathbf{z} = (\mathbf{x}^\top, \mathbf{y}^\top)^\top$, $\omega(\mathbf{z}) = 0$. Thus the Jacobian of $g_d$ at $\mathbf{z}$ becomes

$$\nabla g_d(\mathbf{z}) = I_{n+m} - \alpha(J(\mathbf{z}) + J(\mathbf{z})^\top + \beta(\mathbf{z}))^{-1}$$

$$= \begin{bmatrix} I_n - (2\nabla_{\mathbf{xx}}f + \mathbb{1}_{\{\lambda_{\mathbf{x}}>0\}}(b_{\mathbf{x}})I)^{-1} & 0 \\ 0 & I_m - (-2\nabla_{\mathbf{yy}}f + \mathbb{1}_{\{\lambda_{\mathbf{y}}<0\}}(b_{\mathbf{y}})I)^{-1} \end{bmatrix} \qquad (16)$$

The eigenvalues of $\nabla g_d(\mathbf{z})$ are the eigenvalues of $I_n - (2\nabla_{\mathbf{xx}}f + \mathbb{1}_{\{\lambda_{\mathbf{x}}>0\}}(b_{\mathbf{x}})I)^{-1}$ and $I_m - (-2\nabla_{\mathbf{yy}}f + \mathbb{1}_{\{\lambda_{\mathbf{y}}<0\}}(b_{\mathbf{y}})I)^{-1}$. For an eigenvalue $\lambda$ of $\nabla_{\mathbf{xx}}f$, the corresponding eigenvalue of $I_n - (2\nabla_{\mathbf{xx}}f + \mathbb{1}_{\{\lambda_{\mathbf{x}}>0\}}(b_{\mathbf{x}})I)^{-1}$ will be

$$1 - \frac{\alpha}{2\lambda + \mathbb{1}_{\{\lambda_{\mathbf{x}}>0\}}(b_{\mathbf{x}})}. \qquad (17)$$

If $\lambda_{\mathbf{x}} < 0$, (17) becomes

$$1 - \frac{\alpha}{2\lambda} > 1.$$

As $\mathbf{z}$ is an LASE point, from definition 2.8, $\rho(\nabla g_d(\mathbf{z})) < 1$. Thus, appendix A shows that $\mathbf{z}$ cannot be a LASE if $\lambda_{\mathbf{x}} < 0$. Thus $\mathbf{z}$ is a LASE $\implies \lambda_{\mathbf{x}} > 0 \implies \nabla_{\mathbf{xx}}f \succ 0$. A similar argument by analyzing egeinvalues for $I_m - (-2\nabla_{\mathbf{yy}}f + \mathbb{1}_{\{\lambda_{\mathbf{y}}<0\}}(b_{\mathbf{y}})I)^{-1}$ shows that $\mathbf{z}$ is a LASE $\implies \lambda_{\mathbf{y}} < 0 \implies \nabla_{\mathbf{yy}}f \prec 0$. Thus, from definition 2.2, $z$ is a LASE implies that $\mathbf{z}$ is a strict local Nash equilibrium of (Game 1).

$(2. \impliedby)$ Let $\mathbf{z}$ be a strict local Nash equilbrium. Then, $\lambda_{\mathbf{x}} > 0, \lambda_{\mathbf{y}} < 0$. Clearly, from appendix A, all eigenvalues of $I_n - (2\nabla_{\mathbf{xx}}f + \mathbb{1}_{\{\lambda_{\mathbf{x}}>0\}}(b_{\mathbf{x}})I)^{-1}$ are smaller than 1. Since $\lambda_{\mathbf{x}} > 0, \lambda > 0$. Also, $b_{\mathbf{x}} > \frac{1}{2}, b_{\mathbf{x}} > \frac{\alpha}{2}$, which means that

$$1 - \frac{\alpha}{2\lambda + b_{\mathbf{x}}} > 1 - \frac{\alpha}{2\lambda + \frac{\alpha}{2}} > 1 - \frac{\alpha}{\frac{\alpha}{2}} > -1.$$

Thus $\rho(I_n - (2\nabla_{\mathbf{xx}}f + \mathbb{1}_{\{\lambda_{\mathbf{x}}>0\}}(b_{\mathbf{x}})I)^{-1}) < 1$. Similarly, $\rho(I_m - (-2\nabla_{\mathbf{yy}}f + \mathbb{1}_{\{\lambda_{\mathbf{y}}<0\}}(b_{\mathbf{y}})I)^{-1})$ is less than 1. Thus, from definition 2.8, $\mathbf{z}$ is also a LASE.

(3.) The Jacobian $\nabla g_d$ at any fixed point $\mathbf{z}$ is the same as that given in (16), in which $\nabla g_d$ is clearly symmetric. Thus, $\nabla g_d$ only has real eigenvalues at a fixed point $\mathbf{z}$. $\qquad\square$

**Theorem 3.** *Assume that a strict local Nash equilibrium of Game 1 exists. Under Assumptions 1, 2, 3, and 4, if* DND *converges, it converges to a strict local Nash equilibrium of Game 1. Further, if the step size is chosen as $\alpha_k \leq \max\{2|\lambda_{\mathbf{x}}|, 2|\lambda_{\mathbf{y}}|\}$ then* DND *has a linear local convergence rate of*

$$\lim_{k \to \infty} \frac{\|\mathbf{z}_{k+1} - \mathbf{z}^*\|}{\|\mathbf{z}_k - \mathbf{z}^*\|} \leq \max \left\{ \left(1 - \frac{\alpha}{2\tilde{\lambda}_{\mathbf{x}}}\right), \left(1 + \frac{\alpha}{2\tilde{\lambda}_{\mathbf{y}}}\right) \right\}.$$

*Here, $\alpha$ is the step size at the sequence limit in (10), and $\lambda_{\mathbf{x}}, \lambda_{\mathbf{y}}$ refer to the quantities in (11), and $\tilde{\lambda}_{\mathbf{x}} > 0, \tilde{\lambda}_{\mathbf{x}} < 0$ denote $\lambda_{\mathbf{x}}, \lambda_{\mathbf{y}}$ evaluated at the sequence limit.*

*Proof.* Let $\mathbf{z}^*$ denote the local Nash equilibrium to which DND converges, and let $J^\top J(\mathbf{z})$ denote $J(\mathbf{z})^\top J(\mathbf{z})$. We use Taylor's Theorem (Nocedal and Wright, 1999) applied to $\omega$,

$$\omega(\mathbf{z}_k) - \omega(\mathbf{z}^*) = \int_0^1 J(\mathbf{z}^* + t(\mathbf{z}_k - \mathbf{z}^*))(\mathbf{z}_k - \mathbf{z}^*)dt.$$

For large $k$, as $\mathbf{z}_k \to \mathbf{z}^*$, $J(\mathbf{z}^* + t(\mathbf{z}_k - \mathbf{z}^*)) \approx J(\mathbf{z}_k) \forall t \in [0, 1]$. Also for large $k$, from our assumptions $\beta = 0$ and $E = 0$. Thus we get for large k:

$$
\begin{aligned}
\|\mathbf{z}_{k+1} - \mathbf{z}^*\| &= \|\mathbf{z}_k - \mathbf{z}^* - \alpha_k[J(\mathbf{z}_k)^\top J(\mathbf{z}_k)(J(\mathbf{z}_k) + J(\mathbf{z}_k)^\top)]^{-1}J(\mathbf{z}_k)^\top \omega(\mathbf{z}_k)\| \\
&= \|\mathbf{z}_k - \mathbf{z}^* - \alpha_k(J(\mathbf{z}_k) + J(\mathbf{z}_k)^\top)^{-1}J(\mathbf{z}_k)^{-1}\omega(\mathbf{z}_k)\| \\
&= \|\mathbf{z}_k - \mathbf{z}^* - \alpha_k(J(\mathbf{z}_k) + J(\mathbf{z}_k)^\top)^{-1}J(\mathbf{z}_k)^{-1}(\omega(\mathbf{z}_k) - \omega(\mathbf{z}^*))\| \\
&= \|\mathbf{z}_k - \mathbf{z}^* - \alpha_k(J(\mathbf{z}_k) + J(\mathbf{z}_k)^\top)^{-1}J(\mathbf{z}_k)^{-1}\left(\int_0^1 J(\mathbf{z}^* + t(\mathbf{z}_k - \mathbf{z}^*))(\mathbf{z}_k - \mathbf{z}^*)dt\right)\| \\
&\approx \|\mathbf{z}_k - \mathbf{z}^* - \alpha_k(J(\mathbf{z}_k) + J(\mathbf{z}_k)^\top)^{-1}J(\mathbf{z}_k)^{-1}J(\mathbf{z}_k)(\mathbf{z}_k - \mathbf{z}^*)\| \\
&= \|[I - \alpha_k(J(\mathbf{z}_k) + J(\mathbf{z}_k)^\top)^{-1}](\mathbf{z}_k - \mathbf{z}^*)\| \\
&\leq \|I - \alpha_k(J(\mathbf{z}_k) + J(\mathbf{z}_k)^\top)^{-1}\|_2 \|\mathbf{z}_k - \mathbf{z}^*\|
\end{aligned}
$$

Now, consider the matrix $D_k = I - \alpha_k(J(\mathbf{z}_k) + J(\mathbf{z}_k)^\top)^{-1}$. From the structure of $J(\mathbf{z}_k)$ described in (8),

$$D_k = \begin{bmatrix} I - \frac{\alpha_k}{2}(\nabla_{\mathbf{xx}})^{-1} & 0 \\ 0 & I + \frac{\alpha_k}{2}(\nabla_{\mathbf{yy}})^{-1} \end{bmatrix}.$$

From the properties of $\|\cdot\|_2$ norm,

$$\|D_k\|_2 = \max \left\{ \|I - \frac{\alpha_k}{2}(\nabla_{\mathbf{xx}})^{-1}\|_2, \|I + \frac{\alpha_k}{2}(\nabla_{\mathbf{yy}})^{-1}\|_2 \right\}$$

Let $\lambda_{\mathbf{x}}, \lambda_y$ denote the quantities in (11), evaluated at $\mathbf{z} = \mathbf{z}_k$. Further, let $\tilde{\lambda}_{\mathbf{x}}, \tilde{\lambda}_y$ denote $\lambda_{\mathbf{x}}, \lambda_y$ evaluated at $\lim_{k \to \infty} \mathbf{z}_k$. Then, from Theorem 2, $\tilde{\lambda}_{\mathbf{x}} > 0, \tilde{\lambda}_y < 0$. Thus we can write

$$\lim_{k \to \infty} \|D_k\|_2 = \max \left\{ 1 - \frac{\alpha}{2\tilde{\lambda}_{\mathbf{x}}}, 1 + \frac{\alpha}{2\tilde{\lambda}_y} \right\} < 1 \,\forall\, 0 < \alpha \leq \max\{2|\tilde{\lambda}_x|, 2|\tilde{\lambda}_y|\}$$

Thus,

$$\lim_{k \to \infty} \frac{\|\mathbf{z}_{k+1} - \mathbf{z}^*\|}{\|\mathbf{z}_k - \mathbf{z}^*\|} \leq \lim_{k \to \infty} \|D_k\|_2 < 1$$

This proves that DND has a local linear convergence rate when the step size is chosen as described.

$\square$

**Theorem 4.** *Under Assumptions 1, 2, 3, and 4, $\mathbf{z}$ is a LASE of* SecOND *(Algorithm 1) if and only if $\mathbf{z}$ is a strict local Nash equilibrium of Game 1. Further, assume that a strict local Nash equilibrium of Game 1 exists and let $\mathbf{z}_c$ be the first critical point to which Algorithm 1 comes near to. If $S(\mathbf{z}_k) \approx J(\mathbf{z}_k)^\top J(\mathbf{z}_k)$ (see Appendix D), then Algorithm 1 approaches $\mathbf{z}_c$ superlinearly with a rate*

$$\|\mathbf{z}_{k+1} - \mathbf{z}_c\| \leq L_\omega L_J M \|\mathbf{z}_k - \mathbf{z}_c\|^2, \forall\, k = 0, 1, \dots.$$

*Here $M = \sup_{z \in \tilde{\mathcal{B}}} \|S(\mathbf{z})^{-1}\|$, where $\tilde{\mathcal{B}}$ is the smallest ball centered at $\mathbf{z}_c$ which contains $\mathbf{z}_0$.*

*Furthermore, if Algorithm 1 converges, it converges to a strict local Nash equilibrium of Game 1.*

*Proof.* First, we show that the fixed points of `SecOND` and `DND` are the same. From (12), any fixed point $\mathbf{z}$ of `SecOND` must have $\omega(\mathbf{z}) = 0$, i.e., fixed points $\mathbf{z}$ of algorithm 1 are same as the fixed points of the discrete-time GDA dynamics. Theorem 2 has already established that the fixed points of the discrete GDA dynamics are the same as the fixed points of `DND`.

From (12), when far away from $\mathbf{z}_c$, `SecOND` satisfies the condition that every step is in a feasible descent direction. Further, using a line search rule like (14) ensures that for every step that `SecOND` takes far away from $\mathbf{z}_c$, the merit function $\|\omega(\mathbf{z})\|_2^2$ decreases in value. Thus, when $S(\mathbf{z}_k) \approx J(\mathbf{z}_k)^\top J(\mathbf{z}_k)$, `SecOND` mimics a Gauss-Newton method and from standard nonlinear programming results (Bertsekas, 1997, Proposition 1.1.4), reaches the neighborhood of $\mathbf{z}_c$ superlinearly. Now, when `SecOND` reaches this neighborhood, it changes its dynamics to `DND`, which has already shown to have only local Nash equilibrium points as its LASE points. Clearly, `SecOND` has the same LASE points as `DND` once it switches dynamics, and results from Theorem 2 apply and `SecOND` only converges to a strict local Nash equilibrium. Let us derive the local superlinear rate now. Let $\mathcal{B}_\delta(\mathbf{z}_c)$ denote a ball of radius $\delta$ centered at $\mathbf{z}_c$, and assume that $\mathbf{z}_0 \in \mathcal{B}_\delta(\mathbf{z}_c)$. Let $S(\mathbf{z}_k)$ be denoted by $S_k$. For iteration $k$ when $\|\mathbf{z}_k - \mathbf{z}_{k-1}\| > \epsilon$:

$$
\begin{aligned}
\|\mathbf{z}_{k+1} - \mathbf{z}^*\| &= \|\mathbf{z}_k - S_k^{-1} J(\mathbf{z}_k)^\top \omega(\mathbf{z}_k) - \mathbf{z}^*\| \\
&= \|S_k^{-1}(S_k(\mathbf{z}_k - \mathbf{z}^*) - J(\mathbf{z}_k)^\top \omega(\mathbf{z}_k))\| \\
&= \|S_k^{-1}\left(S_k - J(\mathbf{z}_k)^\top \int_0^1 J(\mathbf{z}^* + t(\mathbf{z}_k - \mathbf{z}^*))dt\right)(\mathbf{z}_k - \mathbf{z}^*)\| \\
&= \|S_k^{-1}\left(\int_0^1 \left[S_k - J(\mathbf{z}_k)^\top J(\mathbf{z}^* + t(\mathbf{z}_k - \mathbf{z}^*))\right]dt\right)(\mathbf{z}_k - \mathbf{z}^*)\| \\
&\le \|S_k^{-1}\|\|\left(\int_0^1 \left[S_k - J(\mathbf{z}_k)^\top J(\mathbf{z}^* + t(\mathbf{z}_k - \mathbf{z}^*))\right]dt\right)\|\|(\mathbf{z}_k - \mathbf{z}^*)\|
\end{aligned}
$$

By choosing $S_k = J(\mathbf{z}_k)^\top J(\mathbf{z}_k)$, and taking $\delta, \epsilon$ to be sufficiently small (and $\epsilon < \delta$), $\|\mathbf{z}_k - \mathbf{z}^*\|$ monotonically decreases and the integral term becomes arbitrarily small for any $k$. Also, due to Assumption 4, $J(\mathbf{z})^\top J(\mathbf{z})$ is a Lipschitz function with a Lipschitz constant of $2L_\omega L_J$, thus from the preceding relation,

$$
\|\mathbf{z}_{k+1} - \mathbf{z}^*\| \le M\left(\int_0^1 2L_\omega L_J t\|\mathbf{z}_k - \mathbf{z}\|dt\right)\|\mathbf{z}_k - \mathbf{z}\| = ML_\omega L_J\|\mathbf{z}_k - \mathbf{z}\|^2
$$

$\square$

**Theorem 5.** *Under Assumptions 1, 2, 3, 4, and 5 hold, and let $\omega(\mathbf{z}) \ne 0 \ \forall \mathbf{z} \in \partial G$. Then, if SeCoND (Algorithm 2) converges to a point $\mathbf{z}$:*

1. *If $\mathbf{z} \in \text{int} G$, then $\mathbf{z}$ is a strict local generalized Nash equilibrium.*

2. *If $\mathbf{z} \in \partial G$, then $\mathbf{z}$ is a local generalized Nash equilibrium (not necessarily strict).*

*Proof.* Assume that `SeCoND` converges to a point $\mathbf{z}$. We consider two cases, as follows:

1. If $\mathbf{z} \in \text{int } \mathcal{G}$, then the immediate neighbourhood around $\mathbf{z}$ which `SeCoND` would have to traverse in order to reach $\mathbf{z}$ is also in $\text{int } \mathcal{G}$. In this neighborhood, the projection step in `SeCoND` does not have any effect, and the algorithm's dynamics follow `DND`. By Theorem 2, `DND` would only have converged to $\mathbf{z}$ if $\nabla f(\mathbf{z}) = 0, \nabla^2_{\mathbf{xx}} f \succ 0$, and $\nabla^2_{\mathbf{yy}} f \prec 0$, which from Definition 2.4 implies that $\mathbf{z}$ is also a strict local Generalized Nash equilibrium.

2. If $\mathbf{z} \in \partial\mathcal{G}$, then from Algorithm 2, $-\omega(\mathbf{z})$ must be in the normal cone of $\mathcal{G}$ at $\mathbf{z}$. Because $\omega(\mathbf{z}) = \begin{bmatrix} \nabla_{\mathbf{x}} f \\ -\nabla_{\mathbf{y}} f \end{bmatrix}$, this means that at $\mathbf{z}$, a feasible step cannot be taken for which $\mathbf{x}$ or $\mathbf{y}$ can reduce or increase $f(\mathbf{x}, \mathbf{y})$, respectively. Thus, from Definition 2.4, $\mathbf{z}$ is a local (not necessarily strict) generalized Nash equilibrium.

This concludes the proof. $\qquad\square$

## B    NOTE ON OUR ASSUMPTIONS

We include this note to give the reader intuition about our Assumptions' validity.

- Assumptions 1, 2, and 4 are standard in the literature (for example, in Adolphs et al. (2019); Mazumdar et al. (2019); Azizian et al. (2024)). Because we propose second-order methods, Assumption 1 ensures that the objective offers meaningful first and second-order derivatives. Assumption 2 ensures that the Jacobians of any dynamical system introduced in the paper can be analyzed at a critical/fixed point.

- In theory, Assumption 3 is required to ensure that all fixed points of the introduced algorithms are critical points of the GDA dynamics (6), and vice versa. Other previous methods have also had to make similar assumptions for this very purpose (Mazumdar et al., 2019), and the assumption we make is easier to check in comparison. The toy and GAN examples in Section 4 do not satisfy Assumption 3, yet we still observe good empirical performance by our proposed approaches.

- **Intuition for Assumption 3:** Consider some smooth function $g(\mathbf{x})$ and the corresponding problem $\min_{\mathbf{x} \in \mathbb{R}^n} g(\mathbf{x})$. Any optimization algorithm will produce iterates of the form $\mathbf{x}_{k+1} \leftarrow \mathbf{x}_k - \alpha_k \mathbf{p}_k$ (or $\dot{\mathbf{x}} = -\mathbf{p}_k$ in continuous-time). In particular, for any Newton-like algorithm, $\mathbf{p}_k = H_k \nabla g(\mathbf{x}_k)$ (for example, $H_k$ can be the regularized hessian inverse $(\nabla^2_{\mathbf{xx}} g(\mathbf{x}_k) + \lambda I)^{-1}$, for some $\lambda \geq 0$). In order to ensure convergence to a minima, one of the conditions developed in nonlinear optimization thoery is that $\mathbf{p}_k$ *must not be orthogonal* to $\nabla g(\mathbf{x}_k)$ when $\nabla g \neq 0$, thus $H_k \nabla g(\mathbf{x}_k)$ *must not be* 0 for $\nabla g(\mathbf{x}_k) \neq 0$. Similarly, in our case, the dynamics (see equations (9), (10)) are iterates of the form $\mathbf{z}_{k+1} \leftarrow \mathbf{z}_k - \alpha_k M_k J(\mathbf{z}_k)^\top \omega(\mathbf{z}_k)$ (or $\dot{\mathbf{z}} = -M(\mathbf{z}) J(\mathbf{z})^\top \omega(\mathbf{z})$), where $M_k$ (or $M(\mathbf{z})$) is a full rank matrix. Thus, to ensure the second term in the update is zero only when $\omega(\mathbf{z}) = 0$, $J(\mathbf{z}_k)^\top \omega(\mathbf{z}_k)$ must not be 0 when $\omega(\mathbf{z}_k) \neq 0$. This directly yields Assumption 3.

- Assumption 5 has been shown to hold for several problems of practical interest (Facchinei and Kanzow, 2010a).

## C    ADDITIONAL EXAMPLE - ENTROPY REGULARIZED ZERO-SUM MATRIX GAME

We consider the following objective function:

$$f(\mathbf{x}, \mathbf{y}) = \mathbf{x}^\top \mathbf{y} - \underbrace{(\mathbf{H}(\mathbf{x}) - \mathbf{H}(\mathbf{y}))}_{\text{entropy regularization}}, \quad \mathbf{x} \in \mathbb{R}^2_+, \ \mathbf{y} \in \mathbb{R}^2_+,$$

where $\mathbf{H}(\mathbf{z}) := \sum_{i=1}^n -\mathbf{z}_i \log(\mathbf{z}_i)$ is the entropy function for some $\mathbf{z} \in \mathbb{R}^n_+$. Based on the above function, we construct the following constrained zero-sum game:

$$\text{Player 1}: \min_{\mathbf{x} \in \mathbb{R}^2_+} f(\mathbf{x}, \mathbf{y}) \qquad \text{Player 2}: \max_{\mathbf{y} \in \mathbb{R}^2_+} f(\mathbf{x}, \mathbf{y}),$$

$$\text{s.t. } \mathbf{x} > \mathbf{0}, \ \mathbf{y} > \mathbf{0},$$

$$\mathbf{1}^\top \mathbf{x} = 1 \text{ and } \mathbf{1}^\top \mathbf{y} = 1.$$

The Nash equilibrium of the above entropy-regularized matrix game is also called the *Quantal Response Equilibrium* (QRE) (McKelvey and Palfrey, 1998). Notably, (i) the above game satisfies Assumption 3, and (ii) the strategies are constrained to lie in the probability simplex, which has an *empty interior*.

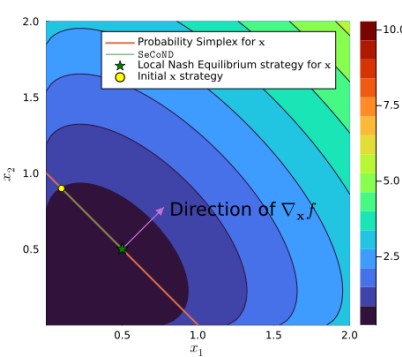 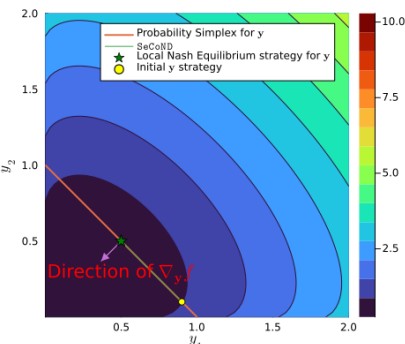

(a) Player 1 ($\mathbf{x}$) initialization at $[0.1, 0.9]^\top$.      (b) Player 2 ($\mathbf{y}$) initialization at $[0.9, 0.1]^\top$.

Figure 3: Numerical results for a constrained game with an empty interior.

**Does `SeCoND` handle cases with empty interiors?** We test `SeCoND` for the above game, and Figure 3 shows the results for both the players. We observe that `SeCoND` successfully reaches the local (generalized) Nash equilibrium located at $\mathbf{x}^* = [0.5, 0.5]^\top$, $\mathbf{y}^* = [0.5, 0.5]^\top$. The probability simplex has no interior, yet the projection scheme in Algorithm 2 ensures that the updates successfully traverse the simplex, reaching the Nash point. At this Nash point, the direction of $\omega$ is such that $\nabla_\mathbf{x} f$ and $\nabla_\mathbf{y} f$ are perpendicular to the directions $\mathbf{x}$ and $\mathbf{y}$ can move in their probability simplexes, thus neither player has any incentive to deviate from this point.

## D   Choice of Regularization Matrices and S in SeCoND

Equation (8) provides a natural choice of $S(\mathbf{z})$ as:

$$S = 2 \begin{bmatrix} (\nabla_{\mathbf{xx}} f)^3 & 0 \\ 0 & -(\nabla_{\mathbf{yy}} f)^3 \end{bmatrix} + \lambda I, \tag{18}$$

where $\lambda \geq 0$ is regularization that ensures positive definiteness. We show how to choose $\lambda$ below.

**Choice of S(z) for superlinear Gauss-Newton Interpretation.** We take $S(\mathbf{z}_k)$ to be $J(\mathbf{z}_k)^\top J(\mathbf{z}_k) + \lambda_k I$ where $\lim_{k\to\infty} \lambda_k = 0$.

A way of designing regularization matrices is by using the Gershgorin Circle Theorem (Horn and Johnson, 2012), which states that for a matrix $A \in \mathbb{C}^{n\times n}$, all eigenvalues of $A$ lie in the union of $n$ discs centred at $A_{ii}$ with radii $R_i = \sum_{j=1, j\neq i}^{j=n} |A_{ij}|$ for $i = 1, \ldots, n$. Thus, to regularize $A$ for invertibility, a diagonal regularization matrix $M$ with the $i^{th}$ diagonal entry $M_{ii} = \mathbb{1}_{\{A_{ii}-R_i<0\}}(|A_{ii} - R_i| + \lambda_0)$, where $\lambda_0 > 0$ is user specified and is a lower bound on the real part of eigenvalues of $A + M$. With this, we design:

1. $E_c(\mathbf{z})$ **in (9):** Here, $A = J(\mathbf{z})^\top J(\mathbf{z})(J(\mathbf{z}) + J(\mathbf{z})^\top)$, and the regularization matrix $E_c(\mathbf{z})_{ii} = \mathbb{1}_{\{A_{ii}-R_i<0 \text{ and } \|\omega(\mathbf{z})\|>\delta_0\}}(|A_{ii} - R_i| + \lambda_0)$. The constant $\delta_0 > 0$ is also user-specified and ensures that at a critical point, $E_c$ is differentiable and that $E_c = 0$.

2. **Design of** $E(\mathbf{z}_k)$ **in (10):** In this case, $A = J(\mathbf{z}_k)^\top J(\mathbf{z}_k)(J(\mathbf{z}_k) + J(\mathbf{z}_k)^\top + \beta(\mathbf{z}_k))$, and we proceed as above.

3. **Design of** $S(\mathbf{z}_k)$ **in (12):** We can take $A$ as the matrix given in Equation (18) and choose $\lambda = \max_i\{(A_{ii} - R_i) + \lambda_0\}$ (and thus $S = A + \lambda I$). For the Gauss-Newton interpretation, we can take $A = J(\mathbf{z}_k)^\top J(\mathbf{z}_k)$.

In our experiments, we took the values $\lambda_0 = 5$ and $\delta_0 = 5 \times 10^{-5}$.

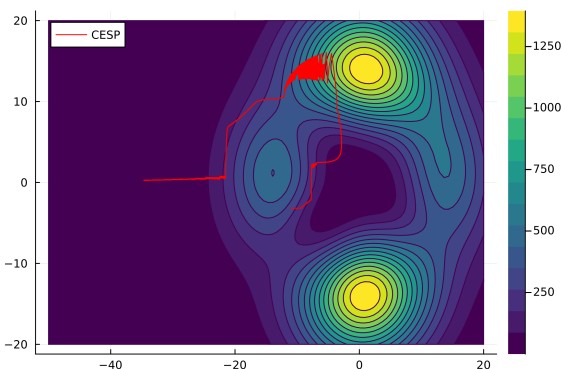

Figure 4: CESP (Adolphs et al., 2019) diverges for the two-dimensional toy example.

## E    EXPERIMENTAL DETAILS

### E.1    TWO-DIMENSIONAL TOY EXAMPLE

#### E.1.1    BASELINES

**Local Symplectic Surgery (LSS).**    For the toy example, the LSS method is:

$$\mathbf{z}_{k+1} = \mathbf{z}_k - \alpha(\omega(\mathbf{z}_k) + e^{-\xi_2\|v\|^2}v),$$

where $v = J(\mathbf{z}_k)^\top(J(\mathbf{z}_k)^\top J(\mathbf{z}_k) + \lambda(\mathbf{z}_k)I)^{-1}J(\mathbf{z}_k)^\top\omega(\mathbf{z}_k)$ and regularization $\lambda(\mathbf{z}_k) = \xi_1(1 - e^{\|\omega(\mathbf{z}_k)\|^2})$. Here, $\xi_1 = \xi_2 = 10^{-4}$. These values have been recommended in the LSS paper for this particular example. Though the authors also described a two-timescale discrete system of LSS, it could not reliably converge for this example, and thus, we resorted to the equation above.

**(Curvature Exploitation for the Saddle Point problem (CESP).**    The CESP method is given by:

$$\mathbf{z}_{k+1} = \mathbf{z}_k - \alpha\omega(\mathbf{z}_k) + \begin{bmatrix} \mathbf{v}_{\mathbf{z}_k}^{(-)} \\ \mathbf{v}_{\mathbf{z}_k}^{(+)} \end{bmatrix},$$

where, for the sign function sgn $:\mathbb{R} \to \{-1, 1\}$,

$$\mathbf{v}_{\mathbf{z}_k}^{(-)} = \mathbb{1}_{\lambda_{\mathbf{x}}<0}\frac{\lambda_{\mathbf{x}}}{2\rho_{\mathbf{x}}}\mathrm{sgn}(\mathbf{v}_{\mathbf{x}}^\top\nabla_{\mathbf{x}}f(\mathbf{x},\mathbf{y}))\mathbf{v}_{\mathbf{x}}$$

$$\mathbf{v}_{\mathbf{z}_k}^{(+)} = \mathbb{1}_{\lambda_{\mathbf{y}}>0}\frac{\lambda_{\mathbf{y}}}{2\rho_{\mathbf{y}}}\mathrm{sgn}(\mathbf{v}_{\mathbf{y}}^\top\nabla_{\mathbf{y}}f(\mathbf{x},\mathbf{y}))\mathbf{v}_{\mathbf{y}}.$$

Here, $\lambda_{\mathbf{x}}$ and $\lambda_{\mathbf{y}}$ denote the minimum and maximum eigenvalues of $\nabla_{\mathbf{xx}}^2 f$ and $\nabla_{\mathbf{yy}}^2 f$ respectively. $\mathbf{v}_{\mathbf{x}}$ and $\mathbf{v}_{\mathbf{y}}$ denote the eigenvectors of $\lambda_{\mathbf{x}}$ and $\lambda_{\mathbf{y}}$. We took $^1\!/_{2\rho_{\mathbf{x}}} = ^1\!/_{2\rho_{\mathbf{y}}} = 0.05$. CESP could not converge reliably for the two-dimensional example, and a typical diverging plot is shown in Figure 4.

#### E.1.2    EXPERIMENT PARAMETERS.

For all algorithms, step size $\alpha$ was taken to be 0.001, except for `SecOND` which performed Armijo line search. Tolerance for convergence was set at $10^{-5}$, and the maximum number of allowable iterations for every algorithm was 15,000. $\epsilon$ for `SecOND` (Algorithm 1) was taken to be $10^{-2}$. For Figure 1, data points that were below $Q_1 - 1.5(Q_3 - Q_1)$ or above $Q_3 + 1.5(Q_3 - Q_1)$ were considered outliers. Here, $Q_1$ and $Q_3$ denote the first and third quartiles, respectively.

#### E.1.3    ADDITIONAL UNCONSTRAINED CASE RESULT.

We show a comparison of `SecOND` and `DND` for the unconstrained toy example to show that our approaches converge to local Nash equilibrium. From Figure 5, it can be seen that only `SecOND` and

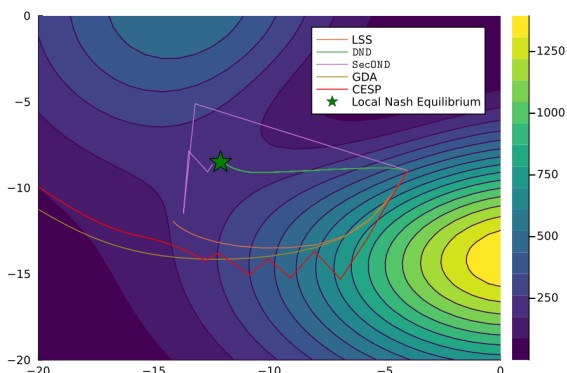

Figure 5: `SecOND` and `DND` converge successfully to a local Nash equilibrium.

`DND` successfully converge to local Nash equilibrium. CESP and GDA diverged, while LSS converged to a non-Nash point. This behavior of LSS arises due to the assumption they make (Theorem 4, (Mazumdar et al., 2019)), which gets violated. Out of the algorithms which converged, LSS took 75 iterations, DND took 5405 iterations, while `SecOND` took just 7 iterations.

### E.2 GENERATIVE ADVERSARIAL NETWORK

#### E.2.1 LSS BASELINE.

For GAN training, we use the two-timescale approximation method for LSS described in (Mazumdar et al., 2019), which is given by

$$\mathbf{z}_{k+1} = \mathbf{z}_k - \gamma_1 \big( \omega(\mathbf{z}_k) + e^{-\xi_2 \|J(\mathbf{z}_k)^\top v_k\|^2} J(\mathbf{z}_k)^\top v_k \big)$$

$$v_{k+1} = v_k - \gamma_2 \big( J(\mathbf{z}_k)^\top J(\mathbf{z}_k) v_k + \lambda(\mathbf{z}_n) v_k - J(\mathbf{z}_k)^\top \omega(\mathbf{z}_k) \big).$$

Similar to the toy example, $\lambda(\mathbf{z}_k) = \xi_1 (1 - e^{\|\omega(\mathbf{z}_k)\|^2})$, and $\xi_1 = \xi_2 = 10^{-4}$ In the generative adversarial network (GAN) example in Section 4.2, the zero-sum game is between the generator $G$, which minimizes $\mathcal{F}$, and the discriminator $D$, which maximizes $\mathcal{F}$. Here, $\mathcal{F} := \mathbb{E}_{x \sim p_{\text{data}}(x)}[\log D(x)] + \mathbb{E}_{\epsilon \sim p_\epsilon(\epsilon)}[\log(1 - D(G(\epsilon)))]$, and $x$ and $\epsilon$ denote actual data samples and noise samples, respectively. Table 1 lists the parameter values of the GAN model used in our evaluation.

Table 1: Parameters of the GAN example in Section 4.2.

|  | Discriminator | Generator |
|---|---|---|
| Input Dimension | 1 | 1 |
| Hidden Layers | 2 | 2 |
| Hidden Units / Layer | 8 | 8 |
| Activation Function | tanh | tanh |
| Output Dimension | 1 | 1 |
| Batch Size | 128 | |
| Dataset size | 10000 | |

We evaluate GDA, LSS, and our `SecOND` approach. GDA uses an Adam optimizer with a learning rate $10^{-4}$; LSS uses an RMSProp optimizer with a learning rate $2 \times 10^{-4}$ for the $x$ and $y$ processes and $1 \times 10^{-5}$ for the $v$ process, as reported in (Mazumdar et al., 2019). `SecOND` uses an RMSProp optimizer with a learning rate $2 \times 10^{-4}$.

**Remark** As suggested by (Goodfellow et al., 2014), to improve the convergence of GDA, we update the discriminator $k = 3$ times more frequent than the generator $G$. Moreover, the GDA generator maximizes $\log(D(G(\epsilon)))$ instead of minimizing $\log(1 - D(G(\epsilon)))$. We found the best practical performance with the said setup.

## F  HARDWARE

The two-dimensional toy examples were run on an Intel i7-11800H 8-core CPU. The GAN training sessions were run on an AMD Ryzen 9 7950X 16-core CPU.

