# OpenReview forum: "Second-Order Algorithms for Finding Local Nash Equilibria in Zero-Sum Games"
_ICLR.cc/2025/Conference — Submitted to ICLR 2025_

### Official Review · Reviewer_mJzP · 2024-11-04

**Soundness:** 3
**Presentation:** 3
**Contribution:** 3
**Rating:** 6
**Confidence:** 3

**Summary:**

The paper considers nonconvex-nonconcave zero-sum games. It provides an algorithm with a local convergence guarantee and (local) linear convergence rate. Further, a variant of that method attains local superlinear convergence to a point that satisfies the first-order Nash equilibrium conditions. Finally, the approach is generalized to handle constraints.

**Strengths:**

The paper considers a well-motivated problem that has attracted much attention in recent years. Compared to prior work that focused on local convergence to Nash equilibria in nonconvex-nonconcave min-max optimization, the paper derives explicit rates of convergence, and also addresses the constrained setting as well. I believe that both of those contributions are important. The authors also support their theory by conducting several experiments, showing that the proposed method outperforms quite convincingly the two most related algorithms (LSS and CESP). Moreover, the paper employs interest techniques, mostly relying on dynamical system theory. Although most of those techniques are standard in this area, the way they are combined in this paper appears to be new. The writing overall is of high quality, and the authors do a good job at explaining the key ideas in the main body. The most related papers have been cited, although I would suggest including a more detailed discussion about the paper of Daskalakis et al. (2023) since it is strongly related. All claims appear to be sound.

**Weaknesses:**

The paper has several weaknesses, some of which are in some sense insurmountable. First, the proposed method relies on second-order information, which could limit its practical applicability. Perhaps more importantly, the main result only establishes local convergence. It is unclear whether a local convergence guarantee is a strong enough solution concept. The paper of Daskalakis et al. (2023) provides a global convergence guarantee, but the latter comes without a rate of convergence. At the same time, I believe that computing local Nash equilibria in this setting is intractable (at least in the constrained setting), and so this perhaps justify focusing on local convergence. Regarding the assumptions made in the paper, Assumption 3 seems very artificial and hard to parse. Can the authors elaborate more on what that assumption entails? And why is it a reasonable assumption? Assumption 2 also seems somewhat artificial, but at least its role is quite clear. I believe that the main body should discuss in more detail some of those assumptions.

**Questions:**

Besides the question above about the underlying assumptions, is it assumed that the set of constraints has a nonempty interior? For example, what if we are dealing with the probability simplex, which has empty interior? It doesn't seem that the definitions and the approach can handle that case.

---

> ### Author Response · Authors · 2024-11-16
> **Response by Authors [1/3]**
>
> We thank the reviewer for their detailed comments. We respond to the questions below. *We are happy to revise our manuscript based on the following discussion/reviewer's suggestions, and have already added an example with empty interior (see Appendix C of the revised manuscript).*
> ### Points raised in the Weakness section.
> * **Local vs global guarantee:** As the reviewer themself point out, the general intractability of the setting is what makes us focus on local (and not global) guarantees. As mentioned in Section 5, working on some global analysis is in the scope of our future work.
> * **Comparison with Daskalakis et al. (2023):** There are two notable differences with the STON'R method introduced in Daskalakis et al. (2023) [A]. These also allow [A] to claim global convergence guarantees, which we currently cannot.
>     1. ***Different solution concept:*** We solve for strict local Nash equilibria (unconstrained case) / generalized local Nash equilibria (constrained case). These are characterized by the first and second-order conditions we mention in Definitions 2.3 and 2.4. On the other hand, STON'R solves for a different solution concept. The min-max critical points that STON'R solves for only have first-order conditions and thus can be viewed as solving for a relaxation of the true local Nash concept. *This means that STON'R can converge to points that do not satisfy Definitions 2.3/2.4.* We would like to emphasize that, as mentioned in Section 3, a big motivation for our work is to find methods that satisfy not only first-order conditions but also second-order conditions.
>     2. ***Constraint Structure:*** The global convergence guarantees of STON'R depend on the constraint set being restricted to a hypercube $(x\in[0,1]^n, y\in[0,1]^m)$, and the authors mention technical challenges in extending the analysis to more general convex constraints (for example, even the unit ball). In comparison, while we cannot offer global convergence guarantees, we do offer at least local convergence in problems with more general constraint structures.
>     3. ***Convergence Rates:*** As pointed out by the reviewer, [A] does not have convergence rates, while we derive explicit convergence rates.
>
> [A] Daskalakis, Constantinos, et al. "Stay-on-the-ridge: Guaranteed convergence to local minimax equilibrium in nonconvex-nonconcave games." The Thirty Sixth Annual Conference on Learning Theory. PMLR, 2023.

---

> > ### Author Response · Authors · 2024-11-16
> > **Response by Authors [2/3]**
> >
> > * **Regarding Assumptions:** (Current discussion in paper is in Appendix B)
> >     1. ***Assumption 3:*** We shall provide some intuition on why Assumption 3 is required, and what it entails via an analogy. Consider some smooth function $g(x)$ and the corresponding problem $\min_{x\in\mathbb{R}^n} g(x)$. Any optimization algorithm will produce iterates of the form $x_{k+1} \gets x_k -\alpha_k p_k$ (or $\dot x = - p_k$ in continuous-time). In particular, for any Newton-like algorithm, $p_k = H_k \nabla g(x_k)$ (for example, $H_k$ can be the regularized hessian inverse $(\nabla_{xx}g(x_k)+\lambda I)^{-1}$, for some $\lambda>0$). In order to ensure convergence to a minima, one of the conditions developed in nonlinear optimization thoery is that $p_k$ *must not be orthogonal* to $\nabla g(x_k)$ when $\nabla g \neq 0$, thus $H_k\nabla g(x_k)$ *must not be* $0$ for $g(x_k)\neq 0$. Similarly, in our case, the dynamics (see equations (9), (10)) are iterates of the form $z_{k+1}\gets z_k - \alpha_k M_k J(z_k)^\top \omega(z_k)$ (or $\dot z = -M J(z)^\top \omega(z)$), where $M_k$ (or $M$) is a full rank matrix. Thus, to ensure the second term in the update is zero only when $\omega(z)=0$, $J(z_k)^\top \omega(z_k)$ must not be $0$ when $\omega(z_k)\neq 0$. We hope this makes the requirement of assumption 3 clear. For it's reasonableness, we would like to highlight that a version of this assumption is standard in related work employing dynamical systems. The closest related state-of-the-art work [B], which uses similar dynamical system concepts, also uses a version of Assumption 3 (in Theorem 4 of [B]). In fact, our Assumption 3 is more relaxed than the corresponding assumption in [B]. Further, from our experiments (which do not satisfy Assumption 3), we empirically see that our method performs well even when Assumption 3 is not met.
> >     2. ***Assumption 2:*** This Assumption is a standard one (for example, in [B], [C], [D]).
> >
> > [B] Mazumdar, Eric V., Michael I. Jordan, and S. Shankar Sastry. "On finding local nash equilibria (and only local nash equilibria) in zero-sum games." arXiv preprint arXiv:1901.00838 (2019).
> >
> > [C] Adolphs, Leonard, et al. "Local saddle point optimization: A curvature exploitation approach." The 22nd International Conference on Artificial Intelligence and Statistics. PMLR, 2019.
> >
> > [D] Azizian, Waïss, et al. "The Rate of Convergence of Bregman Proximal Methods: Local Geometry Versus Regularity Versus Sharpness." SIAM Journal on Optimization 34.3 (2024): 2440-2471.

---

> > > ### Author Response · Authors · 2024-11-16
> > > **Response by Authors [3/3]**
> > >
> > > ### Point raised in Questions Section.
> > > * **Case with empty interior:** We do not assume that the set of constraints must have a non-empty interior. *Our approach can handle the case when the interior is empty*, the local (generalized) Nash concept we find is the one given in Definition 2.4, point 2. The trade-off is that in this case, the point may not be a *strict* local (generalized) Nash equilibrium. The point we converge to in the empty interior is still guaranteed to be a (possibly non-strict) local (generalized) Nash equilibrium because, due to our projection steps in Algorithm 2 (line 372), $\omega$ must be normal to any direction in the constraint set at that point, which satisfies the non-strict sufficiency condition given in Definition 2.4, point 2. Thus, we can handle the cases of empty interior (like that of the probability simplex).
> > > **We have revised the manuscipt and added an example with an empty interior in Appendix C.**
> > >
> > > The example added consists of the objective function $f(\mathbf{x}, \mathbf{y}) = \mathbf{x}^\top\mathbf{y} - (\mathbf{H(x)-H(y)}), \mathbf{x}\in \mathbb{R}^2_+, \mathbf{y} \in \mathbb{R}^2_+$, where $\mathbf{H(\cdot)}$ denotes the entropy function. The game is $\min_{\mathbf{x}\in \mathbb{R}^2_+} f(\mathbf{x}, \mathbf{y})\quad \max_{\mathbf{y}\in \mathbb{R}^2_+} f(\mathbf{x}, \mathbf{y})$ such that $\mathbf{x}> \mathbf{0}, ~ \mathbf{y}>\mathbf{0},~\mathbf{1}^\top\mathbf{x}=1$ and $\mathbf{1}^\top\mathbf{y}=1$. Our method successfully works and finds the local Nash equilibrium located at $\mathbf{x}^* = \mathbf{y}^* = [0.5, 0.5]$.
> > >
> > > We are happy to answer any other questions, and are happy to revise the manuscript to reflect the above discussion as well.

---

> > > > ### Comment · Reviewer_mJzP · 2024-11-25
> > > >
> > > > I thank the authors for the detailed response and the clarifications. I maintain my positive evaluation.

---

### Official Review · Reviewer_S3WW · 2024-11-04

**Soundness:** 2
**Presentation:** 3
**Contribution:** 2
**Rating:** 5
**Confidence:** 3

**Summary:**

This paper tackles the problem of achieving convergence to local Nash equilibria in two-player zero-sum games. While first-order optimization methods struggle with nonconvex-nonconcave settings and often lead to non-Nash equilibria or unstable cycles, this work addresses these challenges by developing algorithms that leverage second-order information to improve stability and convergence specifically towards local Nash equilibria. The author shows that the algorithm achieves local linear convergence and proposes an advanced version enabling superlinear convergence.

**Strengths:**

* The algorithm leverages second-order dynamics to achieve higher convergence rates and avoid common pitfalls like oscillations around equilibrium points.
* The authors validate their approach on a GAN training task, demonstrating the algorithms' robustness and practical utility in adversarial machine learning settings.

**Weaknesses:**

* The explanation of assumptions in Section 3.1 is weak, and the example in the numerical experiment does not satisfy Assumption 3.
* The author does not claim the technique contribution.

**Questions:**

* Theorem 2 does not provide a convergence analysis, is there any technique difficulty?
* It would be better if the author conducts a numerical experiment on a model that satisfies all the assumptions in Section 3.

---

> ### Author Response · Authors · 2024-11-16
> **Response by Authors**
>
> We thank the reviewer for their review. We respond to the reviewer's points below.
> * **Point 1**
> > The explanation of assumptions in Section 3.1 is weak, and the example in the numerical experiment does not satisfy Assumption 3.
>
> Discussion on assumptions is included in Appendix B. We include some further discussion here, and have also revised the manuscript's appendix to reflect the same. Assumptions 1, 2 and 4 are fairly standard in literature, see for example related state-of-the-art methods [A], [B], [C]. Intuition about Assumption 3 can be given through an analogy: consider some smooth function $g(x)$ and the corresponding problem $\min_{x\in\mathbb{R}^n} g(x)$. Any optimization algorithm will produce iterates of the form $x_{k+1} \gets x_k -\alpha_k p_k$ (or $\dot x = - p_k$ in continuous-time). In particular, for any Newton-like algorithm, $p_k = H_k \nabla g(x_k)$ (for example, $H_k$ can be the regularized hessian inverse $(\nabla_{xx}g(x_k)+\lambda I)^{-1}$, for some $\lambda>0$). In order to ensure convergence to a minima, one of the conditions developed in nonlinear optimization thoery is that $p_k$ *must not be orthogonal* to $\nabla g(x_k)$ when $\nabla g \neq 0$, thus $H_k\nabla g(x_k)$ *must not be* $0$ for $\nabla g(x_k)\neq 0$. Similarly, in our case, the dynamics (see equations (9), (10)) are iterates of the form $z_{k+1}\gets z_k - \alpha_k M_k J(z_k)^\top \omega(z_k)$ (or $\dot z = -M J(z)^\top \omega(z)$), where $M_k$ (or $M$) is a full rank matrix. Thus, to ensure the second term in the update is zero only when $\omega(z)=0$, $J(z_k)^\top \omega(z_k)$ must not be $0$ when $\omega(z_k)\neq 0$. We hope this makes the requirement of assumption 3 clear. We would like to re-emphasize that a version of this assumption is standard in related work employing dynamical systems. The closest related state-of-the-art work [A], which uses similar dynamical system concepts, also uses a version of Assumption 3 (in Theorem 4 of [A]). In fact, our Assumption 3 is more relaxed than the corresponding assumption in [A]. **We have added an example that satisfies Assumption 3 in Appendix C. Please refer to the revised manuscript.**
> * **Point 2**
> >The author does not claim the technique contribution.
>
> We are unclear on what the reviewer means here. We will be happy to respond more clearly if the reviewer can elaborate. Our contributions are listed in Section 1. They are:
> 1. **(Unconstrained)** $\texttt{DND}:$ Discrete-time dynamical system that provably converges to only local Nash equilibria (LNE), with a *linear* local convergence rate. In comparison, only two previous related works (LSS and CESP mentioned in the paper) have guarantees of convergence to only LNE, but *do not have any rates in comparison*.
> 2. **(Unconstrained)** $\texttt{SecOND}:$ Discrete-time dynamical system that retains the convergence to only LNE guarantee as above, and converges superlinearly to the neighborhood of a point that satisfies first-order local Nash conditions.
> 3. **(Constrained)** $\texttt{SeCoND}:$ Convergence to only local generalized Nash equilibrium in the presence of (possibly coupled) constraints of the form of a convex set. In contrast, previous work either does not consider this constrained setting (or has a more restrictive constrained settings, like a hypercube) and/or is restricted to the convex-concave case.
>
> * **Point 3**
> >Theorem 2 does not provide a convergence analysis, is there any technique difficulty?
>
> The convergence analysis is the subject of Theorem 3. We apologize for any confusion that has occured. To clarify, Theorem 2 establishes that the only LASE of the proposed dynamics $\texttt{DND}$ are the local Nash equilibrium of the problem (and vice-versa). Thus, if the dynamics converge, they must converge to an LASE (and thus to a local Nash point). The convergence analysis of this is given in Theorem 3.
> * **Point 4**
> > It would be better if the author conducts a numerical experiment on a model that satisfies all the assumptions in Section 3.
>
> *We have revised the manuscipt and added an example that satisfies Assumption 3 in **Appendix C**, and our method performs successfully for that example as well.*  The experiments currently in Section 4 were chosen so as to investigate whether the method empirically performs well even if Assumption 3 is violated, which it does.
>
> [A] Mazumdar, Eric V., Michael I. Jordan, and S. Shankar Sastry. "On finding local nash equilibria (and only local nash equilibria) in zero-sum games." arXiv preprint arXiv:1901.00838 (2019).
>
> [B] Adolphs, Leonard, et al. "Local saddle point optimization: A curvature exploitation approach." The 22nd International Conference on Artificial Intelligence and Statistics. PMLR, 2019.
>
> [C] Azizian, Waïss, et al. "The Rate of Convergence of Bregman Proximal Methods: Local Geometry Versus Regularity Versus Sharpness." SIAM Journal on Optimization 34.3 (2024): 2440-2471.

---

### Official Review · Reviewer_CwFG · 2024-11-04

**Soundness:** 3
**Presentation:** 2
**Contribution:** 3
**Rating:** 6
**Confidence:** 3

**Summary:**

One of the fundamental problems in algorithmic game theory area is learning algorithms for games. In this paper, the authors study a problem related to this area, particularly an unconstrained two-player zero-sum game with an objective function that can be nonconvex-nonconcave.

In this direction they study algorithms for convergence to local Nash equilibria taking into account the second order conditions of the function. To do that they propose a dynamical systems approach s.t. under specific assumptions the critical points are also local Nash equilibria to the initial game. They prove that their initial dynamics converges with local linear rate of convergence to a local strict Nash equilibrium and they give an improved algorithm of them with super linear local rate of convergence. Finally, they give an algorithm for the constrained setting based on an Euclidean projection approach to converge to a local generalized Nash equilibrium and experiments to give evidence for their approach.

**Strengths:**

I think that the problem that the authors study is significant and this paper gives new directions to understand it. Their results are good since they give new dynamics and prove local rate of convergence using techniques such as bounding by strictly less than 1 the eigenvalues of the Jacobian matrix. Furthermore, the idea to find a dynamical system s.t. any LASE is a local Nash equilibrium of the initial game is nice.

**Weaknesses:**

The results hold under specific assumptions and they only guarantee local convergence. I think the paper needs more elaboration how strict are these assumptions and for the latter how far the initial point should be at most from the LASE point in order to have convergence.

**Questions:**

Does the assumptions imply unique LASE point?
Is there any assumption on the uniqueness of the local strict Nash equilibrium?
How far should the initial point be at most from the LASE point in order to have convergence?
Do the proven rates of convergence of the unconstrained case hold in the constrained case?

---

> ### Author Response · Authors · 2024-11-16
> **Response by Authors**
>
> We thank the reviewer for their detailed comments. Please find our discussion on the weaknesses and answers to the questions below.
> ### Regarding Weaknesses
> * **Strictness of these assumptions:** Assumptions 1, 2, and 4 are standard in the literature (for example, [A], [B], [C]). Assumption 5 has been shown to hold for several problems of practical interest [D]. Assumption 3 is required to ensure that all fixed points of the introduced algorithms are critical points of the GDA dynamics (given in equation (6)), and vice-versa. A version of assumption 3 is also used in related work employing dynamical system concepts. The closest related state-of-the-art work [A], which uses similar dynamical system concepts, also uses a version of Assumption 3 (in Theorem 4 of [A]). In fact, our Assumption 3 is more relaxed than the corresponding assumption in [A]. While relaxing our Assumption 3 is scope of future work, we have deliberately chosen our examples in Section 4 such that Assumption 3 does not hold in order to demonstrate that our methods perform well empirically even without Assumption 3. We refer the reviewer to Appendix B where we discuss all assumptions, and also to our response to reviewer XyrN for a more detailed discussion on Assumption 3. The manuscript has been revised to reflect these discussions in Appendix B.
>
> ### Response to Questions
> * **Do the assumptions imply unique LASE point?** No, in a problem that satisfies all assumptions, there can be multiple LASE points (and hence multiple local Nash equilibria).
> * **Is there any assumption on the uniqueness of the local strict Nash equilibrium?** No, there is no assumption on uniqueness of local strict Nash equilibrium. Our convergence guarantees are for convergence to *a* LASE/local Nash point, so depending on different initializations, one can go to different LASE/local Nash points.
> *  **How far should the initial point be at most from the LASE point in order to have convergence?** In general, the initial point must be within the region of attraction of at least one LASE point. In nonconvex settings, there is no clear answer for the extent of this region for most local methods. A potential direction of future work consists of constructing Lyapunov functions for the introduced algorithms which can help establish the extent of the region of attraction. However, we note two important points in our current methods: 1) The initial point need not be in a locally convex-concave region, 2) The modified Gauss-Newton method in $\texttt{SecOND}$ (Algorithm 1) can reach a critical point with the initialization arbitrarily far away from it, provided any standard line search method developed in nonlinear optimization (for example, Armijo) is done.
> *  **Do the proven rates of convergence of the unconstrained case hold in the constrained case?** No, this is because at the boundary, the gradients of active constraints can be arbitrarily aligned with the update step. Thus we have been unable to conduct a meaningful analysis of the convergence rate in the constrained nonconvex-nonconcave setting. To the best of our knowledge, related work is also unable to handle this case. The only results we are aware of pertain to the convex-concave setting (for example, [E]).
>
> We are happy to answer any other questions, and are happy to revise the manuscript to reflect the above discussion as well.
>
> [A] Mazumdar, Eric V., Michael I. Jordan, and S. Shankar Sastry. "On finding local nash equilibria (and only local nash equilibria) in zero-sum games." arXiv preprint arXiv:1901.00838 (2019).
>
> [B] Adolphs, Leonard, et al. "Local saddle point optimization: A curvature exploitation approach." The 22nd International Conference on Artificial Intelligence and Statistics. PMLR, 2019.
>
> [C] Azizian, Waïss, et al. "The Rate of Convergence of Bregman Proximal Methods: Local Geometry Versus Regularity Versus Sharpness." SIAM Journal on Optimization 34.3 (2024): 2440-2471.
>
> [D] Facchinei, Francisco, and Christian Kanzow. "Generalized Nash equilibrium problems." Annals of Operations Research 175.1 (2010): 177-211.
>
> [E] Cevher, V., Piliouras, G., Sim, R., & Skoulakis, S. (2023). Min-max optimization made simple: Approximating the proximal point method via contraction maps. In Symposium on Simplicity in Algorithms (SOSA) (pp. 192-206). Society for Industrial and Applied Mathematics.

---

> > ### Comment · Reviewer_CwFG · 2024-11-26
> > **Response to the Authors**
> >
> > Thank you very much for your complete and analytical answers. I will keep my initial score on the review of the paper.

---

### Official Review · Reviewer_XyrN · 2024-11-05

**Soundness:** 3
**Presentation:** 3
**Contribution:** 3
**Rating:** 8
**Confidence:** 3

**Summary:**

The authors design a (family of) second-order algorithms for two-player zero-sum games with the property that, if convergence is achieved, the point of convergence is proved to be a local Nash equilibrium. In particular, such a point of convergence must be a locally asymptotically stable equilibrium.

**Strengths:**

It's noteworthy that the authors developed an entire family of algorithms for, both, unconstrained and constrained spaces.

**Weaknesses:**

1. An obvious weakness is that the suggested algorithms are applicable only to two-player zero-sum games. Could they be modified to support multi-player games?
2. Assumptions 1, 2,  and 4 seem reasonable. I am not quite sure how common Assumption 3 is. I understand that the authors consider the relaxation of Assumption 3 as future work. Could they provide some possible intuition? For example, is their natural subclass of two-player zero-sum games that this assumption is satisfied?

**Questions:**

Kindly refer to the weaknesses.

---

> ### Author Response · Authors · 2024-11-16
> **Response by Authors**
>
> We thank the reviewer for their detailed comments. Kindly find our answers below:
>
> **Question on more than two players:** The reviewer is correct in pointing out that the current method is only applicable to the two-player case. The reason why the current method cannot be directly applied to a case of N>2 players is that our second order dynamics rely on a particular structure induced by the two players. This particular structure can be lost in the more general case of N>2 players. This being said, the two-player case is important in itself, as many problems of interest across computer science, economics, control theory, etc. have a two player zero-sum formulation. Thus, we (and most previous related works also) focus on the two-player setting.
>
> **Question on Assumption 3:**
> * For **intuition**, we provide an analogy with single player optimization. Consider some smooth function $g(x)$ and the corresponding problem $\min_{x\in\mathbb{R}^n} g(x)$. Any optimization algorithm will produce iterates of the form $x_{k+1} \gets x_k -\alpha_k p_k$ (or $\dot x = - p_k$ in continuous-time). In particular, for any Newton-like algorithm, $p_k = H_k \nabla g(x_k)$ (for example, $H_k$ can be the regularized hessian inverse $(\nabla^2_{xx}g(x_k)+\lambda I)^{-1}$, for some $\lambda\geq0$). In order to ensure convergence to a minima, one of the conditions developed in nonlinear optimization theory is that $p_k$ *must not be orthogonal* to $\nabla g(x_k)$ when $\nabla g \neq 0$, thus $H_k\nabla g(x_k)$ *must not be* $0$ for $\nabla g(x_k)\neq 0$ [A, Chapter 3]. Similarly, in our case, the dynamics (see equations (9), (10)) are iterates of the form $z_{k+1}\gets z_k - \alpha_k M_k J(z_k)^\top \omega(z_k)$ (or $\dot z = -M J(z)^\top \omega(z)$), where $M_k$ (or $M$) is a full rank matrix. Thus, to ensure the second term in the update is zero only when $\omega(z)=0$, $J(z_k)^\top \omega(z_k)$ must not be $0$ when $\omega(z_k)\neq 0$. We hope this makes the requirement of assumption 3 clear. We would like to highlight that a version of this assumption is standard in related work employing dynamical systems. The closest related state-of-the-art work [B], which uses similar dynamical system concepts, also uses a version of Assumption 3 (in Theorem 4 of [B]). In fact, our Assumption 3 is more relaxed than the corresponding assumption in [B].
> * An *example of a sub-class of games* where this assumption holds is games where the matrix $J$ is non-degenerate. In fact, a popular class of games that satisfy Assumption 3 are entropy-regularized zero-sum matrix games. The Nash equilibria of such games are called *Quantal Response Equilibria*, which have important applications [C]. **We have revised our manuscript to include an example that uses our method to solve such a game, in Appendix C. We have also revised our discussion of Assumptions in Appendix B to include the above intuition.**
>
> We are happy to answer any other questions, and are happy to revise the manuscript to reflect the above discussion as well.
>
> [A] Nocedal, Jorge, and Stephen J. Wright, eds. Numerical optimization. New York, NY: Springer New York, 1999.
>
> [B] Mazumdar, Eric V., Michael I. Jordan, and S. Shankar Sastry. "On finding local nash equilibria (and only local nash equilibria) in zero-sum games." arXiv preprint arXiv:1901.00838 (2019).
>
> [C] McKelvey, Richard D., and Thomas R. Palfrey. "Quantal response equilibria for normal form games." Games and economic behavior 10.1 (1995): 6-38.

---

### Author Response · Authors · 2024-11-16
**Summary of revisions to manuscript**

We have made the following changes to our submission based on the reviewers' feedback:
* Added more discussion about our assumptions, especially intuition about assumption 3 in Appendix B (as requested by multiple reviewers).
* Added an example that satisfies assumption 3 (as requested by reviewer S3WW) and has an empty interior (as requested by reviewer mJzP) in Appendix C.

We are happy to make more changes based on further reviewer feedback.

---

### Meta-Review · Area_Chair_ezQb · 2024-12-19

**Metareview:**

This paper examines and analyzes a second-order algorithm for learning in non-convex/non-concave min-max games. The algorithm under study can be viewed as a Newton-like variant of an earlier algorithm proposed by Mazumdar et al. (2019), and the authors show that, if the algorithm converges, it converges to a strict local Nash equilibrium, and it does so at a superlinear rate.

This paper received one clear accept recommendation but, otherwise, most of the reviewers were on the fence. The concerns raised during the discussion mostly focused on the following issues:
1. It was not clear what role Assumption 3 played in the analysis. A similar assumption did appear in a previous paper by Mazumdar et al. (2019), but it did not seem to play the same role there (in the sense that it is not needed for results that are similar to those of the current paper).
2. There were also concerns regarding the positioning of the current paper relative to the work of Mazumdar et al. (2019). In particular, the dynamics considerd in this paper are very similar to the LSS dynamics of Mazumdar et al., but the authors did not adequately discuss the differences between them (either in the paper or during the discussion phase).
3. There are also issues parsing the authors' results as stated. To name but an example, Point (1) of Theorem 2 states that "*the fixed points of DND are *exactly* the fixed points of the discrete-time GDA dynamics*". This means that DND admits stationary points that are critical points of the game, but not necessarily local Nash equilibria. However, if DND is initialized at such a point (or, more generally, at the stable manifold of such a point), it will converge to it, in violation of the authors' claim in L266 of Theorem 3.
4. Similar concerns apply to the statement "$S(\mathbf{z}_k) \approx J(\mathbf{z}_k)^\top J(\mathbf{z}_k)$" in Theorem 4 and the like: despite references to the paper's appendix, the exact meaning of the "approximate equality" sign $\approx$ can only be guessed, and likewise for the quantifier "near" in the same statement: for example, if DND starts at a non-Nash critical point, it will always stay there, contradicting the authors' claim.
5. The computational cost of the proposed algorithm is not adequately discussed. For instance, knowing the functional dependence of one's payoff function on one's individual control variables is quite different from knowing this dependence on the other player's control variables; likewise, assuming access to individual gradients or Hessians - that is, derivates of one's payoff function relative to one's individual control variables - is different than assuming access to gradients and/or Hessians relative to the other player's control variables; and again so for the Armijo-like line search in (14), etc.

After discussing these concerns, it was not possible to make a clear case for accepting the paper "as is" - i.e., without further revisions that would require a fresh set of reviews - and it was decided to make a reject recommendation. This decision was made with the best interests of the paper in mind: the paper is not without its merits, but the current version is not clear on several key issues that end up  undermining the authors' contributions, so a more drastic revision would be in order before the paper could ultimately be accepted.

**Additional Comments On Reviewer Discussion:**

In regard to the concerns raised above, Reviewer XyrN was more positive during the discussion phase. However, further discussion showed that these concerns were deeper than initially thought, hence the conclusion that the paper cannot be accepted at this stage (at least, not without a fresh round of reviews).

---

### Decision · Program_Chairs · 2025-01-22

Reject